# The Use of Hydraulic Fracturing in Stimulation of the Oil and Gas Wells in Romania

**Ion Pană** [1] , **Iuliana Veronica Gheţiu** [2], **Ioana Gabriela Stan** [2,*], **Florinel Dinu** [2], **Gheorghe Brănoiu** [3] **and Silvian Suditu** [2]

1. Mechanical Engineering Department, Petroleum-Gas University from Ploieşti, 100680 Ploieşti, Romania; ion.pana@upg-ploiesti.ro
2. Well Drilling, Extraction and Transport of Hydrocarbons Department, Petroleum-Gas University from Ploieşti, 100680 Ploieşti, Romania; ivghetiu@gmail.com (I.V.G.); flgdinu@yahoo.com (F.D.); silviusuditu@yahoo.com (S.S.)
3. Petroleum Geology and Reservoir Engineering Department, Petroleum-Gas University from Ploieşti, 100680 Ploieşti, Romania; gbranoiu@yahoo.com
* Correspondence: gabriela.stan@upg-ploiesti.ro

**Abstract:** This paper presents the application of the hydraulic fracturing method in Romania, exemplified by three case studies. In the current conditions in which the oil and gas prices have risen above the limit of affordability, Romania, one of the few producers in Europe, is trying to solve the problems that have arisen through various methods, which are as follows: offshore drilling, gas underground storage, field rehabilitation and increasing the efficiency of applied technologies. The application of hydraulic fracturing is a safe process, with minimal environmental implications and certain economic benefits. The important thing is to have the necessary energy now, in the desired quantities and with minimal expenses. The authors sought to include key issues in the application of this technology in Romania. The scientific literature on this topic has helped us to interpret the data from the field in difficult situations and were a real support in our activity. We need to provide energy support and energy security and we do not have a lot of resources. Under these conditions, the reactivation of existing deposits and the extension of the production period are essential elements. The authors designed the fracturing technologies. The data corresponding to the geological structure obtained through geological investigations, and the database corresponding to the analyzed wells from the company's data archive were the elements used in the simulation programs. Thus, the values in the fracturing area about pore fluid permeability, layers stress, Young's modulus of the structure and fracture toughness were established. The fluids for the fracturing operation and the proppant were chosen for each case, in accordance with the geological recommendations, by our team. Testing of the fracturing technologies for different variants of the pumping program was carried out using the Fracpro program. The variants presented in this article are some of the best solutions found. We used the step-by-step flow test to find the fracture expansion pressure and closing pressure for each case. The mini-frac program established corrections to the designed technologies during the operation quickly and with reduced costs. The designed technologies allowed us to anticipate the necessary flows and pressure, leading to the choice of equipment. The fracture operations were performed only after the projected technologies anticipated the economic benefits covering the investments for the use of the equipment and the operation itself. Knowing the measured pressure of the well and the conditions of communication with the gas/oil reservoir, a simulation of the gas/oil production that could be obtained was made with the simulator. Two situations were exemplified for a gas well and an oil well. The field production results for a two-year interval are also indicated for these wells and a comparison was made with the estimated production.

**Keywords:** hydraulic fracturing; fracture geometry; stimulation operation; fracturing fluid; fracturing technology

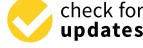



## 1. Introduction

Hydraulic fracturing of a layer means the creation of cracks or the opening and expansion of natural cracks, by pumping into the layer a fluid composed of water, sand, and chemicals with high pressure, which exceeds the strength of rocks in the area surrounding the wellbore. Hydraulic cracking is a physical process in which the layer yields on the planes of minimum resistance, under the effect of the pressure of the fluid pumped into the well. The purpose of the hydraulic crack treatment is to increase the flow of fluid in the layer area around the borehole by a certain radius, equivalent to the radius of the crack created. These treatments are applied in the following situations: to layers consisting of consolidated rocks, sandstones, limestones, dolomites and conglomerates with low permeability to increase the inflow of crude oil/natural gas/thermal waters; to the injection wells, to increase the receptivity of the layers; for the successful cementation of aquifers. By creating a crack in the layer, changes occur both in the flow system by the distribution of current lines and in the shape of the pressure variation curve around the borehole.

The differences between the physical and social sciences literature on fracking (and between the two and the coverage of fracking in the popular media) are striking. So far, the physical science literature suggests that the effects of fracking on the environment are not as dramatic as they are frequently described; more importantly, it appears that many of the most worrying issues can be addressed through improved oversight/regulation and the use of the existing and emerging best management practices. This is in stark contrast to the almost apocalyptic descriptions of environmental and social degradation presented in the most critical social science literature [1].

The article [2] addresses hydraulic fracturing for shale gas extraction as a matter of interpretive policy. Gathering empirical cases from several countries, the following three approaches to the interpretation of hydraulic fracturing are identified: understanding its meaning, contextual explanation of the institutionalization of its meaning, and designing policies as an intervention to change its meaning. The following two central tensions have been identified in the case of shale gas in all cases: (1) economic opportunity or threat to the environment and (2) transition to a carbon-free energy future or the perpetuation of a fossil fuel system. The answer to these two problems has generated certain public policies at the government level.

The global growth of renewable energy initiatives has been followed by the need to include the social impact of any project as a core element. Significant challenges for renewable energy development include uncertainty in assessing social impact at the local level, participation, and social acceptance [3].

People and organizations campaigning against hydraulic fracturing technology are grouped around impact headlines. When browsing the Internet, these areas have increased accessibility. Hopke and Simis [4] discuss the discourse on hydraulic fracturing on the social networking platform Twitter in a period of intense public disputes over the applicability of the technology. They studied the relative importance of negative messages about shale gas development in relation to pro-shale messages on Twitter on five hashtags, concluding that the public is more interested in the idea of dangerous technology and not the scientific aspects. Finding the segmentation between the interested audience involved in the speech—the hashtag audience—has significant implications for both policymakers and the idea of online deliberation. Politicians who use new media platforms to measure public opinion will receive very different opinions, depending on the hashtags they follow.

Evidently, when there are issues related to the harmful effects of the technology, they need to be told. The role of the disclosure of damage is to generate actions to eliminate the shortcomings and not to block technology with promising results. This approach is highlighted by the author of [5]. Directional drilling and hydraulic fracturing technologies dramatically increase natural gas extraction. There is systematic evidence for methane contamination of drinking water associated with shale gas extraction in overlapping aquifers over shale formations. Regulations are needed to ensure a sustainable future for shale gas extraction and to improve public confidence in its use. The paper [6] analyses the

statistical situation of induced earthquakes that have occurred since 1929, at magnitudes equal to or greater than 1.0. The causes of induced earthquakes are mining; depletion of the oil and gas field; water injection for secondary oil recovery; water accumulation in the reservoir; waste disposal; academic research drilling investigating seismicity; mining drilling; geothermal operations and hydraulic fracturing for the recovery of gas and oil from sedimentary rocks with low permeability. The earthquakes triggered by the last cause had magnitudes between 1 and 3.8. To date, hydraulic fracturing has been a relatively benign mechanism compared to other anthropogenic triggers, probably due to the low fluid volumes and short pumping times used in hydraulic fracturing operations. The analysis is intended to inform stakeholders about hydraulic fracturing technology.

Countries' global efforts are focused on reducing carbon emissions and reducing global warming. Sustainable development issues, reiterated at Glasgow COP 26, are very current. The results confirm a global interest in renewable and unconventional resources for the coming decades, as well as efforts to increase the potential for energy efficiency in all end-use sectors to reduce the overall environmental impact of the production of energy [7].

Mixed energy solutions that provide a stable source of energy, combined with alternative wind, solar and renewable sources, are currently a transitional solution. A conventional ground-based heat pump can be used to supplement/reduce heat, creating a hybrid system that is cost-effective for a particular climate with large temperature variations [8]. Study [9] proposes an optimal functioning of the coordinated distribution of gas and electricity networks, by taking into account the interconnected energy nodes. The proposed energy hub is equipped with combined heat and power units, a boiler, energy storage, a heat pump, and a gas unit to meet the heating and electric load requirements.

The energy efficiency strategy is integrated into the business model of the companies. The companies have significant energy savings in the exploration and production segments through technical and organizational measures. An example is the improvement of the energy efficiency of the gas extraction process elevator presented by the author of [10].

From an economic point of view, the global transition to green energy is difficult, and on the road to this goal, improving operating technologies is a viable medium. Uncertainty about the production of renewable sources undermines the reliability of the energy system, requiring additional spare capacity. Several case studies are presented in [11–14]. The study [15] estimates the costs induced by an additional spare capacity to reduce the uncertainty of solar generation in the Korean energy system and analyzes the effectiveness of the energy storage system in reducing these costs. Obviously, new renewable energy technologies are creating jobs. The study [16] assesses the effects of regional clean energy production in identifying jobs generated in the renewable energy sector. Romania is one of the major producers of oil and gas in Europe. The mining sector has been considerably reduced in recent years, contributing to Romania's effort to integrate into the joint effort to ensure optimal living conditions for the current and future generations. The wind sector, solar panels, and nuclear energy are an important segment of the Romanian energy market. Energy operators' perspectives on energy transitions need to be systematically included in the discourse on the sustainable development of renewable energies. It has been shown that the challenges and problems that arise in the context of regional energy transformation cannot be generalized beyond regional circumstances; rather, they should be considered as specific regional phenomena that need to be overcome through regionally adapted energy concepts [17]. The study [18] compares the impact of economic incentives on attitudes towards the acceptance of fossil fuels, renewable energy, and nuclear energy.

It is necessary to create a national framework to ensure the transition to a renewable energy economy. The paper [19] focuses on developing a framework for overcoming the existing barriers and facilitating sustainable and carbon-neutral technologies at the national level. The Romanian national natural gas transmission company started the activity for the introduction of the hydrogen transmission system, generated by its own factories. It is estimated to be 10% by 2030. Renewable hydrogen production is an excellent method of storing and transferring energy from intermittent renewable sources, such as wind and

sun, so that it can be used at our choice [20]. Energy companies are interested in bringing low-cost gas and oil to the market by capitalizing on old deposits. The recent rise in energy prices has created a chain of price increases, which has led to a melting of the population's savings, and a decrease in the actual wage. Many businesses that relied on the use of large amounts of energy have stopped. The overlapping effects of the COVID-19 crisis can lead to the accumulation of frustrations for the population. With existing potential, the application of a new technology that has proven to be very efficient can currently provide an additional source of energy at the national level to overcome an internal energy crisis, coupled with other necessary measures.

## 2. Bibliographic Research and Systematization of Knowledge on Hydraulic Fracturing

### 2.1. Bibliographic Study

The knowledge of the technical problems of hydraulic fracturing is based on reading important books in the field.

Ref. [21] addresses the theoretical background of one of the most widespread activities in hydrocarbon wells, that of hydraulic fracturing and basic aspects, such as elasticity, stress distribution, fluid flow, and the dynamics of the rupture process in terms of the influence of those phenomena on the fracture created are included.

Big companies developed manuals for workers [22], with the intention to be used by junior engineers who want to gain some knowledge about the fracturing process and by experienced engineers who want to gain a deeper perspective in certain areas.

In recent years, the increase in the volume of knowledge has led to the appearance of many books with a large quantity of pages [23,24]. Some of these focus on the economic problems of hydraulic fracturing [25].

Evidently, there is a need to use software products that allow the study of complicated problems that arise with the use of this technology. Shen and Standifird present in [26] a 3D numerical modeling technology for hydraulic fracturing developed in recent years and introduce solutions to various 3D geomechanical problems related to hydraulic fracturing.

Several companies have developed such applications and a good review of these can be found in the paper [27]. For a better understanding of the sensitivities of the hydraulic fracturing modeling process to different boundary conditions and input parameters, the Environment Agency asked the British Geological Survey (BGS) to review the available software used by the oil and gas industry to model fracturing hydraulics. In summary, these products are MFrac (Baker Hughes, Houston, TX, USA), FracMan (Golder Associates, Mississauga, ON, Canada), Kinetix Stimulation (Schlumberger, Houston, TX, USA), GOHFER (Barree & Associates, Lakewood, CO, USA), StimPlan (NSI Technologies, Tulsa, OK, USA), Fracpro (Carbo Ceramics, Houston, TX, USA), and FrackOptima (FrackOptima, Riverside, CA, USA). It is evident that there are not many, but the support for their development is quite strong, as demonstrated below.

### 2.2. Fluid Characteristics for Hydraulic Fracturing

The fluids used must satisfy the following conditions: good stability to temperature and pressure variations; adequate viscosity support to transport the material layer cracks; not to react with minerals in collecting rocks or fluids that saturate these rocks, resulting in insoluble compounds; not to form emulsions in the layer; the possibility of removal from the layer without difficulties, after the completion operation. The article [28] reviews both the traditional viscous fluids used in conventional hydraulic fracturing operations and the new family of fluids, polymers, developed for both traditional and unconventional reservoirs. Polymers have played a major role in the success of the hydraulic fracturing process. The review article [29] aims to provide an overview of the polymeric systems currently used as a backing and coating agent. Large amounts of water used in hydraulic fracturing can constrain the production of oil and natural gas in shale lands. The study [30] looked at the amount of fresh water and recycled water used to fracture wells from 2008 to 2014 in several oil fields in the United States. The results showed that the average annual

volumes of water used per well ranged from 1000 m$^3$ to 30,000 m$^3$. The percentage of water used for hydraulic fracturing in each state was relatively low compared to the use of water for other industries. From 2009 to 2014, 6.55% of the volume of water used in hydraulic fracturing contained recycled water or recycled hydraulic fracturing wastewater. Recycled water was used in 10.84% of the hydraulic fracturing wells.

### 2.3. Data on the In Situ Fracturing Operation

Data on the history of wells, the hydraulic fracturing technologies used, the effects on production, the duration of the stimulus effect, the economic and environmental aspects, and the success of methods for predicting results through software products are rarely present in the scientific literature. In the section dedicated to case studies, the authors of this paper will indicate three examples of how to solve these problems. In this paper, we mention pulse hydraulic fracturing (PHF), a new stimulation technique to effectively improve the permeability of coal seams, increasing the efficiency of gas extraction [31]. However, the mechanism of PHF is still unclear, especially the effects of key parameters. In this study, a series of experiments using a 3D load were performed to investigate the effect of pulse rate on fracture propagation and extension during PHF. The results indicated that pressure changes during PHF can be divided into the following three periods: rising, plateau, and decreasing, and plateau is the key period. In addition, the lower frequency corresponded to a smoother pressure curve with a longer period. In this case, the sample was "softened", the pressure to initiate the fracture was lowered and more complicated fractures could be created. A higher frequency corresponded to a fluctuating pressure curve with a shorter period. In this case, the fractures can easily form due to the rapid increase in pressure.

### 2.4. Experimental Laboratory Data on Hydraulic Fractures

Experiments on hydraulic fracturing with shale, sandstone, and granite cores are presented in [32–36]. The use of an X-ray scanner and tomographic computer scan show an increase in rock permeability and the results provide good results for the use of hydraulic technologies [32–34]. In addition, hydraulic fracturing is a promising simulation technique, used in improving geothermal systems to increase the heat production of a geothermal reservoir by improving the permeability of the rock of the reservoir and hydraulic fracturing experiments, performed under 3D stress at 20 °C, 100 °C, 200 °C, 300 °C, and 400 °C, showed that the cooling effect of the fracturing fluid for high-temperature drilling can lead to thermal shock and cause tensile stresses near the surface of the borehole [35]. Hydraulic fracturing of granite cores shows that rock temperature is one of the most important factors that affects hydraulic fracturing pressure [36].

### 2.5. Software and Fracture Simulation

As we mentioned earlier, the software products need strong support from laboratory experiments to improve the accuracy of solving practical applications, related to hydraulic fracturing. The purpose of the studies [37,38] was to investigate the effect of turbulent fluid flow on the planar propagation of a hydraulic fracture. The modeling of a hydraulic fracture includes solving the following: the equation of elasticity that ensures the balance of the rock, the equation of equilibrium of the volume of the fluid, and the equation of fluid flow, which are solved together with a propagation condition. In these papers, the influence of turbulent flow is condensed into a single friction factor that influences the fluid flow equation, i.e., the relationship between fluid flow and pressure gradient. The results for a plane fracture that propagates in a three-layer geometry demonstrate that turbulence leads to a more circular fracture that promotes height increase through an area of 59 high stress.

The aim of this comprehensive study [39] is to conduct an investigation into the influence of five important criteria on hydraulic fracturing techniques and the selection of the best technology to improve the oil recovery factor. Of the five fundamental criteria for the selection and operation of hydraulic fracturing, the in situ stress-strain criterion is the

most important. The effect of in situ stress-strain on hydraulic fracturing procedures has played a substantial role in creating hydraulic fractures in several locations and can control the rate of propagation. The effect of rock heterogeneity on hydraulic fracturing procedures related to the variation in reservoir characteristics, especially rock properties involving porosity, permeability, and Young's modulus, has had a profound impact on hydraulic fracturing procedures. Therefore, the fluctuation of these parameters may be affected by the location and datasets of the selected area. The formations initially have fractures and cracks from the initial base of the formation, most of which are close to the fractures created during the process. Therefore, these natural fractures can significantly influence the fracturing process, due to the connectivity induced during hydraulic fracturing. The aim of the study [40] is to develop a model for the transport of the support agent in hydraulic fractures that takes into account the gravitational and shielding effects. The model blocks particle access from reaching the crest points by imposing a width restriction based on particle size. The introduced equations model the propagation of hydraulic fractures and the transport of the supporting agent inside them. They are based on the equations of the constant flow of a viscous fluid, mixed with spherical particles, in a channel. This transport model applies to the two fracture geometries, Khristianovich–Zheltov–Geertsma–De Klerk (KGD) and pseudo 3D (P3D). Numerical simulations show that the proposed method makes it possible to highlight the formation and growth of plugs, as well as the gravitational placement of the support agent for both geometries.

Hydraulic fracturing in permeable rock is a complicated process that can be influenced by a variety of factors, including operational parameters (e.g., fluid viscosity, injection rate, and hole diameter) and in situ conditions (e.g., stress and initial level of stress, pore pressure). In order to elucidate the effects of these variables, simulations are performed on laboratory scale samples using the fully coupled discrete element (DEM) method. The model is validated by comparing the stress around the drilling wall measured with that theoretically calculated [41]. A shale reservoir is one of the largest unconventional resources and has obvious anisotropic characteristics, due to its inherent sedimentary structures. The viscosity and flow capacity of the fracturing fluid plays an important role in the process of hydraulic fracturing. Shale hydraulic fracturing tests were performed using freshwater and supercritical $CO_2$ as fracturing fluids to investigate different ways of propagating fractures. The orientation of the shale layer had a profound influence on the propagation of the fracture when used in either freshwater or supercritical $CO_2$ [42]. The rupture pressure of the shale core was also affected by the fracturing fluid. Macroscopic observation of the fractures revealed different fracture geometries and propagation patterns. Anisotropic structures and fracturing fluids have been shown to influence the path of hydraulic fracturing.

Compact limestone formations are difficult to stimulate by hydraulic fracturing, due to their low porosity, low permeability, and high density. The fracture geometry is neither a two-winged fracture found in a homogeneous sandstone formation nor a complex network of fractures that occur in a shale formation with well-developed discontinuities and high fragility. In the study [43], a series of laboratory simulated experiments were performed to investigate the behavior of fracture initiation and propagation in the formation of compact limestone. The results showed that a complex network of fractures hardly formed in compact limestone. The fractures initiated in the section of open holes or perforations propagate in four forms, resulting in the following three main categories of fracture geometry: transverse fracture, longitudinal fracture, and complex fracture. Low fluid viscosity or low pump speed has been shown to be an effective method for activating and opening discontinuities, forming complex fractures under a small difference in horizontal stress, rather than a large difference in horizontal stress. In addition, an acidic fluid must be injected into the limestone formation before the fracturing operation to reduce the strength of the rock, resulting in a low breaking pressure. Stimulation technology by hydraulic fracturing is an effective method for increasing the production of methane in the coal layer, especially for the coal layer with low permeability, low pressure in the reservoir, and low gas saturation.

Based on the theories and methods of oil geology and the mechanics of flow through porous media, the paper [44] presents a two-phase model, 3D flow, and double-porosity hydraulic fracturing. The model was successfully applied in a coal methane reservoir. The actual gas production data are consistent with the results calculated by the new model. The models related to hydraulic fracturing require continuous improvement. Experimental investigations are a valuable support. The studies [45,46] present the acquisition, analysis, and interpretation of data on acoustic emissions from a series of laboratory experiments of hydraulic fracturing on granite.

Articles [47–51] describe the importance of 2D and 3D modeling of hydraulic fracturing. The variation in the injection pressure, the type of rock, and the cracking fluid used are the defining elements that must be taken into account.

Furthermore, a 3D test system was designed for the experiments that simulate the hydraulic fracturing of the horizontal shale outcrops in [52]. The effects of fracture fluid viscosity and flow rate on fracture propagation can be expressed by the product of these two parameters. The experiment results show that when the order of magnitude of $q \cdot \mu$ is $10^{-9}$ Nm, it is favorable to the creation of the fracture network, but both too low and too high $q \cdot \mu$ values are harmful.

The phase-field method applied to solid fracturing has been incorporated into porous media theory to describe the hydraulic fracturing scenarios. This method has been found to be very convenient, not only for the description of solid deformation and fracture but also for the transition of Darcy-type flow in saturated porous media to Navier–Stokes flow in fractured areas. However, due to the monotonous evolution of the phase-field variable, the approach to the fracture in the phase-field does not usually allow the description of pre-existing closed fractures or fractures that close after generation. The study [53] refers to the introduction of a crack opening indicator as an additional variable governed by the existing deformation. By using this procedure, not only the opening but also the closing of the fractures, as well as the pre-fractured domains, can be easily included in the numerical simulation of the fracking scenarios in saturated porous media.

The paper [54] presents a new pseudo-3D model for a hydraulic fracture that grows into a stratified rock, with differences in both material properties and in situ stresses. In this model, the vertical plane of the fracture is divided along the lateral direction into the cells. Within each cell, the cross-sectional deformation is a flat deformation, and the fluid pressure is allowed to vary vertically. The fluid flow in the cell can act in two directions. Along the central part, which is of uniform pressure, the fluid flow is lateral, corresponding to the main component of the fluid transport. Near the vertical fracture edge of a cell, the flow can be vertical and is generated by the vertical pressure gradient. This part of the cell is called the filling part. When the pressure on the filling side reaches a level equal to that on the center, the flow direction changes from vertical to lateral. Both the filling and the central parts increase the fracture height.

The article [55] presents a framework for the numerical modeling of hydraulic fracture in dense fragile rocks with low permeability. Numerical results are compared with experimental data. The study uses a biphasic material consisting of a solid phase and a fluid. The behaviors, such as solid deformation and fluid-pore flow, are described using the theory of the continuum of extended porous media. Hydraulically operated cracking is modeled in the sense of brittle fracture, using a procedure based on minimizing energy. In order to calibrate the model, 2D and 3D simulations of the problems with the initial limit values are introduced and compared with the laboratory experiments on hydraulic fracturing. The article's contributions can be implemented in finite element simulation models, providing a robust solution to hydraulic fracturing problems.

The directional hydraulic fracturing technique (DHF) is part of China's national energy strategy to increase the permeability of the large-scale coal seam in coal mines to exploit the methane in the coal seam. The study [56] tests the law of variation in crack propagation and initiation pressure, through physical simulation experiments. The results show that the DHF technique can achieve the oriented propagation of fractures along the desired

direction. The maximum main stress, the fracture extension area, and the change of the fracture direction on the three axes are analyzed by the FLAC3D numerical software. The hydraulic fracturing directional mechanism is well-founded, which provides good experimental support for theoretical research and subsequent engineering applications.

In papers [57,58], we have a new approach for hydraulic fracturing known as the CZM model. The Mohr–Coulomb criterion is used to reveal the rock fracture behavior in numerical simulators.

There are a lot of factors that affect the treatment of hydraulic fracturing, i.e., in situ forming stresses, fracture fluid properties, support agent, pumping velocity, reservoir fluid, rock properties, etc. For predictive modeling, these factors are associated with a lot of uncertainties, as most are measured in the laboratory, calculated, or subjectively estimated. Moreover, the precise contribution of each factor to the final fracture result is unknown for each individual case. Therefore, for better treatment performance and to find the best range of design parameters, a predictive model of hydraulic fracturing involving these uncertainties is necessary, especially for the newly operated shale gas reservoir. In paper [59], a new uncertainty-based approach is chosen for hydraulic fracturing processes. It is based on assigning the probability distribution for some variables and parameters used in the design process. These probability distributions are used as input data for analytical equations that describe fracturing processes. The Monte Carlo simulation technique is used to apply uncertainty-based values to analytical design formulas. A hypothetical example of hydraulic fracturing is used to simulate the effect of various variables and design parameters on the entire fracturing process.

### 3. Case Studies and Results

In this section, we present three case studies from three wells. We designed the article in a practical way, in which hydraulic fracturing works can be performed. A case study means a blocked or low production well that is trying to get back to production at better values.

We know the structure of the deposit, the construction of the well, the exploitation history, and the characteristics of the extracted product. These are the data of the problem.

We rely heavily on a simulator to design and verify hydraulic fracturing technology. There are not many recognized products in the field and they have different prices, including Fracpro (this product was used), Saphire, Resfrac, Gohfer, and Petrel. Theoretical aspects, research in the field, and limitations of theoretical models are useful for those who produce this computer software. These aspects must also be known by those who apply them in order to appreciate the degree of confidence in the predictions of the product used.

Using the simulator requires a good geological engineer in the team. We cannot start over with geological investigations. We save time and money using structural experience, complemented by some geological measurements. The simulators require a lot of data that we need to provide. In reality, all the data cannot be of a very good quality.

The second important issue concerns the fluids used and the proppant. The use requires some experience in the field and a specialist to interpret the information that appears. Compatibility with the structure has to be ensured. Costs are important. We indicated the characteristics of the flow tests to establish the expansion and closure pressure of the fracture. The values of the fracture extension presssion and fracture closing pressure were estabilised for each case with a step-by-step flow variation test.

The next aspect is related to equipment. Working with a particular company often involves adapting to its equipment. The numerical simulation orients us on them because the treatment stages generate the necessary pressure conditions and flows, as demonstrated by the figures with the pumping diagrams. However, as mentioned above, we are conditioned by certain equipment and we need to change the technology to adapt to it.

The analysis of designed fracturing technologies requires a good knowledge of the theory in the field, so here we can effectively use this information. Establishing a route of communication with the deposit is based on many factors that need to be interpreted

correctly. The success of the technology is appreciated by the following simulator diagrams: proppant concentration, fracture conductivity, proppant volume fraction, slurry temperature, slurry concentration, acid concentration, reservoir etching, proppant location, growth rings, fluid positions, etc. The progress made here is spectacular, as stated in the introductory section. The technologies that gave good results in the three cases are presented in this article. A good result means production close to the estimated design.

During the design of the fracturing technology, the mini-frac operation performed before the main fracturing treatment allows the following to be established: after fracturing pressure falls off; post-shut-in analysis; after closure analysis; closure parameters; flow regimes; reservoir pressure; far filed transmissibility. Mini-frac tests are included in the article.

Simulators also offer the possibility of predicting production, which is a good measure to cover the costs of fracturing. Two cost analyses were performed for a gas well (Case 1) and for an oil well (Case 3). The results are positive in both situations, with a plus for the natural gas well. The benefits are also accentuated by the recent rise in prices.

In the two analyzed cases, comparisons are made with the production that was recorded after the fracturing operation. Sometimes, the production is weaker than anticipated and the blocking of the well or the abrupt decrease in the production after a period of time are problems that we face.

The first well is located in the Padina structure (Figure 1), which is a faulted anticline, trending WSW-ENE. The structure is divided into the following two blocks: western and eastern. The eastern block contains hydrocarbons in the Albian, Senonian, Sarmatian, and Meotian.

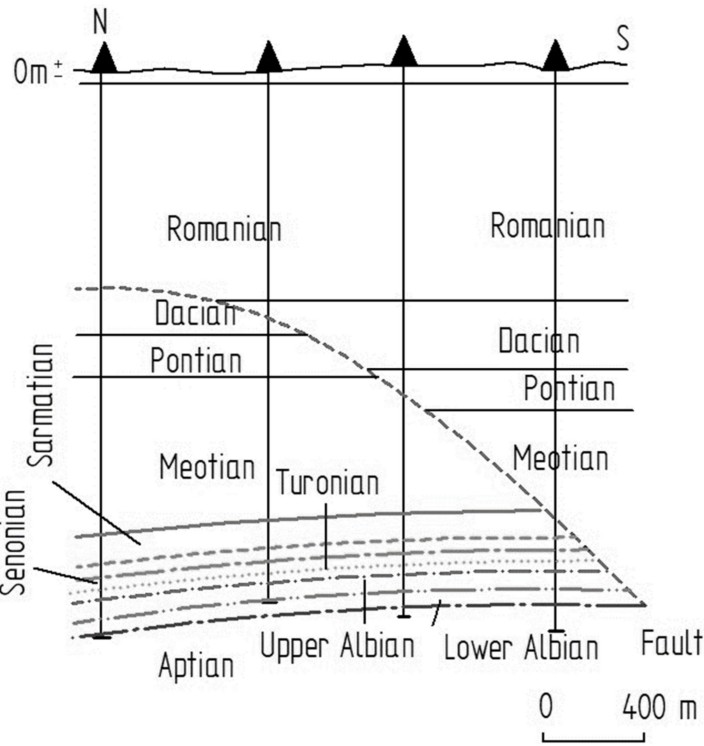

**Figure 1.** The geological cross-section of Padina structure.

The Meotian contains gas in several sand packets. In these deposits, the gas is not uniformly distributed due to the structural element, as well as the lithostratigraphic features.

The Meotian developed in the molasse facies consists of sandy marls alternating with marly sands or weak consolidated sands, gray-yellow, micaceous, with fine-medium particle size. The lithological constitution of the Meotian is not uniform and the Meotian presents strong variations in lithofacies. It has thicknesses between 550–600 m.

The production of the Meotian free gas reservoir is considered finished, as the wells stopped production due to flooding, and digging new wells is not economically viable.

The second well is located in the Bradu structure, which belongs to the regional scale of the Eastern Getic Basin. The Bradu structure (Figure 2) is a faulted anticline, with an oil reservoir in a stratiform structural trap type. The wells drilled on the structure were crossed geological formations belonging to Romanian, Dacian, Pontian, Meotian, and Sarmatian, discordantly arranged over Moesian Platform deposits. The main productive formation on the structure is the Meotian, which is characterized by an arenite sandy facies with 50–100 m thickness and is represented in the northern part by the Me1+2 complex layers and in the southern part by Me1+2 (in the bottom) and Me2 complexes (in the top). The Meotian reservoir, located at a depth of about 1250 m, is composed of weakly consolidated clay sands with a high content of lutites (bigger than 25%). The crude oil is non-paraffinic with 23 API degrees. The average daily oil production is around 17–18 $\frac{m^3}{day}$. The Meotian deposits have the following average values of main parameters: porosity 23–26%; permeability 108 mD; irreducible water saturation 25%; volume reduction factor 1.08; crude oil viscosity 19 cP; solution oil ratio 30–35 $\frac{Sm^3}{m^3}$.

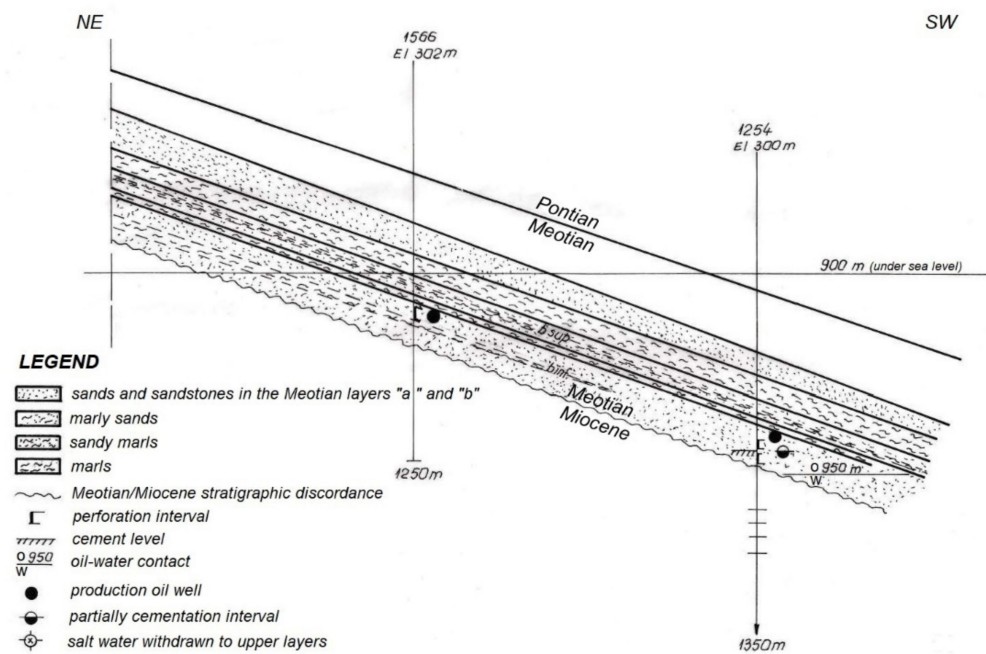

**Figure 2.** The geological cross-section of Bradu structure (no scale).

The third well is located on the Moinesti structure. The accumulations of hydrocarbons on the Moinesti structure, which is analyzed in this paper, are found in deposits of the Kliwa sandstone. The traps are of structural type, both in the lower part (Oligocene I) and in the upper one (Oligocene II), and generally include crude oil deposits, with dissolved gas, in blocks MO I, MO II, MO V-IX, MO VI-IX, and gas oil fields, in areas MO II and MO IV, at the transition horizon.

At the level of the transition horizon, multi-layered deposits with marginal water and with an average saturated oil thickness of about 63 m were found, and at supraKliwa, the deposits are of massive water type.

Regarding the physical parameters of the reservoir rock, they were determined by statistically processing the values obtained from the analysis of the cores extracted by samples from the productive formation. The Kliwa sandstone, with a maximum development in this area (250 m), is composed of intercalations of siliceous sandstone of Kliwa type, with a thickness of 1–10 m, disodilic schists, thin intercalations of siliceous sands, and hard siliceous conglomerates, which are gray with green schists elements.

The supra Kliwa structure, represented by intercalations of disodiles and Kliwa sandstones, develops in thicknesses varying between 30–80 m. Due to some zonal developments, the separation of this structure from the previous one is difficult to achieve and this is why in all the works it was admitted that it forms a single hydrodynamic unit exploiting together.

### 3.1. Case 1. PADINA Well

The operation of stimulation aims to increase the productivity index of the well, applying a sustained hydraulic stimulation treatment. The fluid used is polymer-based. The deposit is of the following Meotian Vb type: the average temperature in the deposit is 39 °C, porosity 21%, permeability 60 mD, initial saturation in water 40% and relative natural gas density extracted from the well is 0.56. The geometric elements of the gas well, casing, tubing, and the location and dimensions of the perforations are given in Figure 3.

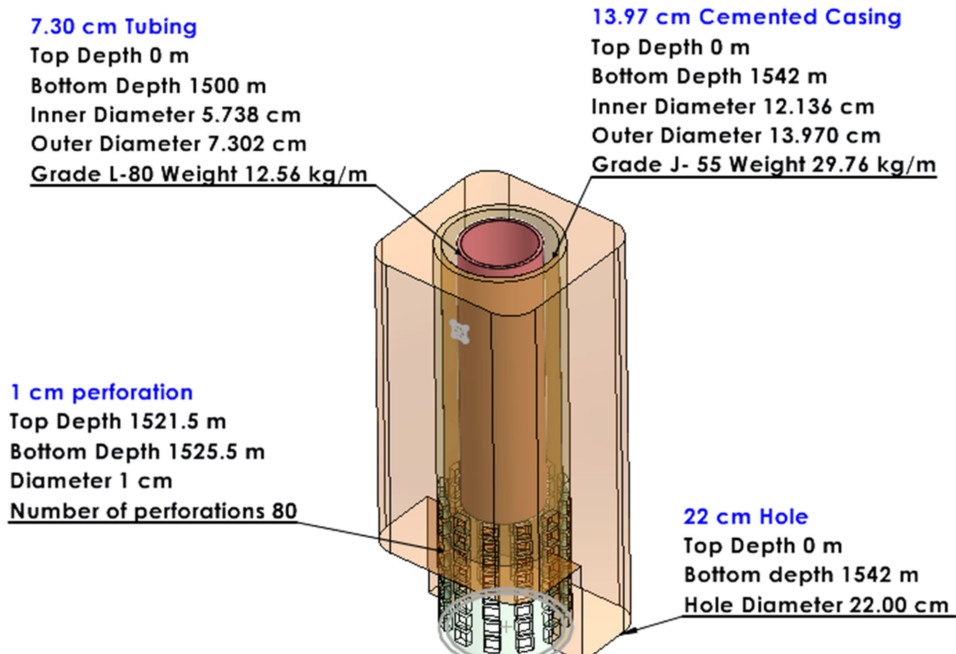

**Figure 3.** Geometric elements of the gas well, casing, tubing, and perforations. Case 1.

The choice of fracturing fluid is made while taking into account the following criteria: the filtrate of the fluid must be small enough to allow the extension of the fracture along the designed length; it has to ensure a low blockage. The fluid HG 40 was chosen. The support material was chosen by the following criteria: its strength should be higher than the closing pressure of the fracture; to have a high permeability at the fracture closing pressure, over 1350 mD · m. The chosen support material (proppant) was Jordan Sand 20/40.

The hydraulic fracturing program includes the following steps:

a.　Preparation of the operating fluid, consisting of filtered water from the reservoir, with 3% KCl;
b.　Arming the packer and performing flowing tests with the operating fluid;
c.　Preparing of 3 m$^3$ of acid solution;
d.　Preparing the cracking fluid;
e.　The test of variation in the flow in steps up and down, Table 1;
f.　The analysis of the mini-frac program, Table 2;
g.　Initiation and extension of the fracture;
h.　Filters packaging.

**Table 1.** Pumping program for step-by-step flow variation test (linear fluid). Case 1.

| Cycle | Stage | Injection Flow | Injection Time | Planned Volume | Total Planned Volume |
|---|---|---|---|---|---|
| | | m$^3$/min | s | m$^3$ | m$^3$ |
| | Rate 1 | 0.20 | 60 | 0.20 | 0.20 |
| | Rate 2 | 0.30 | 60 | 0.30 | 0.50 |
| | Rate 3 | 0.40 | 60 | 0.40 | 0.90 |
| | Rate 4 | 0.50 | 30 | 0.25 | 1.15 |
| Step up | Rate 5 | 0.60 | 30 | 0.30 | 1.45 |
| | Rate 6 | 0.70 | 30 | 0.35 | 1.80 |
| | Rate 7 | 0.90 | 30 | 0.45 | 2.25 |
| | Rate 8 | 1.10 | 30 | 0.55 | 2.80 |
| | Rate 9 | 1.30 | 30 | 0.65 | 3.45 |
| | Rate 10 | 1.50 | 30 | 0.75 | 4.20 |
| | Rate 11 | 1.20 | 20 | 0.50 | 4.70 |
| Step down | Rate 12 | 1.00 | 20 | 0.33 | 5.03 |
| | Rate 13 | 0.50 | 20 | 0.16 | 5.19 |

**Table 2.** Mini-frac pumping program (linear fluid).

| Stage | Planned Volume | Proppant Concentration | Injection Flow |
|---|---|---|---|
| | m$^3$ | $\frac{kg}{m^3}$ | m$^3$/min |
| Filling of the tubing | 5.00 | 0 | 1.50 |
| Pad | 10.00 | 0 | 1.50 |
| Prop slug | 5.00 | 200 | 1.50 |
| Pad | 10.00 | 0 | 1.50 |
| Relocation | 5.00 | 0 | 1.50 |
| Shut-in | 0 | 0 | 0 |

The step-by-step flow variation test is performed before the hydraulic fracturing to determine the following values: the fracture closure pressure, the fracture closure pressure, and the pressure losses near the perforations.

The operating fluid is injected into the formation, increasing the flow in steps, as demonstrated in Table 1. The bottom pressure is recorded as a function of the injection flow. The typical appearance is two segments, which form an obtuse angle. At the point where the curve changes its slope, the fracture extension pressure (FEP) is determined. The value of FEP for this case was 275 bar. The intersection between the straight line drawn by regression through the points of the curve after the change of slope and the axis OY gives an approximate value of the closing pressure of the fracture (FCP). The value of FCP was 223.75 bar. After the fracture is initiated, the flow rate is decreased gradually, and the pressure is measured. After the shutting down of the pump, the instant shut-in pressure (ISIP) is recorded. The calculated bottom pressure, minus the ISIP for all flow values, expresses the friction losses near the wellbore through the perforations, as an injection flow function. The regression analysis determines the pressure losses near the well, depending on the injection flow. The interpretation is to what extent the pressure losses depend on perforations and tortuosity. The characteristics of the rock zones traversed by the natural gas well in the fracturing zone are indicated in Table 3.

The mini-frac analysis is performed based on the data in Table 2. After pumping the fluid to the mini-frac, the closure of the fracture is expected and the pressure drop is recorded. After analyzing the pressure data, the operation is redesigned with the new data. The initiation and extension of the fracture are carried out according to the program indicated in Table 4 and Figure 4a. One of the most important goals of stimulation is to obtain fractures with high conductivity. Fracture conductivity measures the total flow rate through a unit length of a fracture and is calculated as the product of fracture permeability and fracture width. The presence of proppant in the fracture is also a measure of the

quality of the fracture, as demonstrated in Figure 4b. The amount of proppant placed in the fracture is measured by proppant concentration, defined as the proppant mass per unit fracture-face area and usually in the unit of $\frac{kg}{m^2}$. A large amount of proppant also has economic implications, increasing the cost of fracturing. Critical conductivity is assessed as the conductivity reaching a production of 97% of the estimated one. The assessment of critical conductivity can be made depending on the sustained length of the fracture and the years/months of production. In Figure 5, such an assessment is indicated qualitatively [60]. For the present article, the examples made aimed at ensuring uniform conductivity of the fracture and over an area as wide as possible.

**Table 3.** The characteristics of the rock zones traversed by natural gas well in the fracturing zone. Case 1.

| Depth TVD | Depth MD | Layer Thickness | Rock Type | Pore Fluid Permeability | Leak-Off Coefficient | Stress | Young's Modulus | Poisson's Ration | Fracture Toughness |
|---|---|---|---|---|---|---|---|---|---|
| m | m | m | - | mD | $m \cdot min^{1/2}$ | bar | MPa | - | $MPa \cdot m/s$ |
| 891 | 891 | 599 | Shale | 0 | 0 | 213.6 | 17,200 | 0.25 | 0.22 |
| 1480 | 1480 | 14 | Shale | 0 | 0 | 269.1 | 17,200 | 0.25 | 0.22 |
| 1494 | 1494 | 10 | Sandstone | 60 | 1530 | 237.4 | 17,200 | 0.20 | 0.11 |
| 1504 | 1504 | 8 | Shale | 0 | 0 | 272.9 | 17,200 | 0.25 | 0.22 |
| 1512 | 1512 | 2 | Sandstone | 60 | 1530 | 239.6 | 13,800 | 0.20 | 0.11 |
| 1514 | 1514 | 7 | Shale | 0 | 0 | 274.6 | 17,200 | 0.25 | 0.22 |
| 1521 | 1521 | 8 | Sandstone | 60 | 1530 | 241.5 | 13,800 | 0.20 | 0.11 |
| 1529 | 1529 | 8 | Shale | 0 | 0 | 277.5 | 17,200 | 0.25 | 0.22 |
| 1538 | 1538 | 10 | Sandstone | 60 | 1530 | 244.3 | 13,800 | 0.20 | 0.11 |
| 1548 | 1548 | 0 | Sandstone | 60 | 1530 | 245.1 | 13,800 | 0.20 | 0.11 |

**Table 4.** The hydraulic fracturing treatment from the Fracpro simulator. Case 1.

| Stage Type | Flow Rate | Proppant Concentration | Clean Volume | Stage Length | Cumul Time | Fluid Type | Proppant Type |
|---|---|---|---|---|---|---|---|
| | $m^3/min$ | g/L | $m^3$ | min | min:s | | |
| Main frac pad | 1.50 | 0 | 25.00 | 16.67 | 16:39 | HG_40 | |
| Main frac slurry | 1.50 | 100 | 5.00 | 3.46 | 20:07 | HG_40 | Jordan Sand 20/40 |
| Main frac slurry | 1.50 | 300 | 5.00 | 3.71 | 23:50 | HG_40 | Jordan Sand 20/40 |
| Main frac slurry | 1.50 | 400 | 5.00 | 3.84 | 27:40 | HG_40 | Jordan Sand 20/40 |
| Main frac slurry | 1.50 | 500 | 5.00 | 3.96 | 31:38 | HG_40 | Jordan Sand 20/40 |
| Main frac slurry | 1.50 | 600 | 9.00 | 7.36 | 38:59 | HG_40 | Jordan Sand 20/40 |
| Main frac slurry | 1.50 | 700 | 14.00 | 11.80 | 50:47 | HG_40 | Jordan Sand 20/40 |
| Main frac slurry | 1.50 | 800 | 10.00 | 8.68 | 59:28 | HG_40 | Jordan Sand 20/40 |
| Main frac slurry | 1.50 | 900 | 12.00 | 10.72 | 70:11 | HG_40 | Jordan Sand 20/40 |
| Shut-in | 0.00 | 0 | 0.00 | 4.00 | 74:11 | Shut-in | |

The data used in the Fracpro simulator were as follows: *for rocks*, (Table 3 and Figure 6b,d), the measured permeability and porosity, stress, Young's module, Poison's ratio, fracture toughness from the geological information of the drilling companies; *for well*, (Figure 3), site information; *for treatment*, (Table 4 and Figure 4), it was designed by the authors and corrected from the step-by-step flow and mini-frac tests, (Tables 1 and 2) from field data. The use of the Fracpro program allows us to anticipate the size of the fracture, the concentration of proppant in the fracture, and its conductivity. The fracturing parameters at 1521.5 m (Figure 6a,c and the application report) are half from the fracture length, 14.6 m; fracture height at the end of the operation, 31.8 m; half of the sustained width of the fracture, 7.7 m; sustained fracture height, 16.8 m; the upper limit of the created

fracture, 1510.4 m; the lower limit of the created fracture, 1542.2 m; the depth of the fracture supported at the top, 1513 m; the depth of the fracture supported at the base, 1534 m; net pressure in the fracture at the end of the operation, 87.5 bar; average surface pressure, 239 bar; average conductivity of the fracture, 2374.8 mD · m; dimensionless conductivity of the fracture, 5.14; fracture closing time, 74.11 min; maximum fracture width, 8.05 cm; average fracture width, 5.04 cm; average proppant concentration in the fracture, 22.69 $\frac{kg}{m^2}$; average hydraulic power, 726 kW; total proppant pumped, 31,475 kg; total proppant in the fracture, 16,520 kg. The logging diagram of the well is shown in Figure 6b,d. It is used to determine the permeability of the geological layers. From the pressure variation diagram, Figure 6e, a significant increase in the pressure is observed after the initiation of the fracture. The growth of the fracture continues until the end of the planned operations.

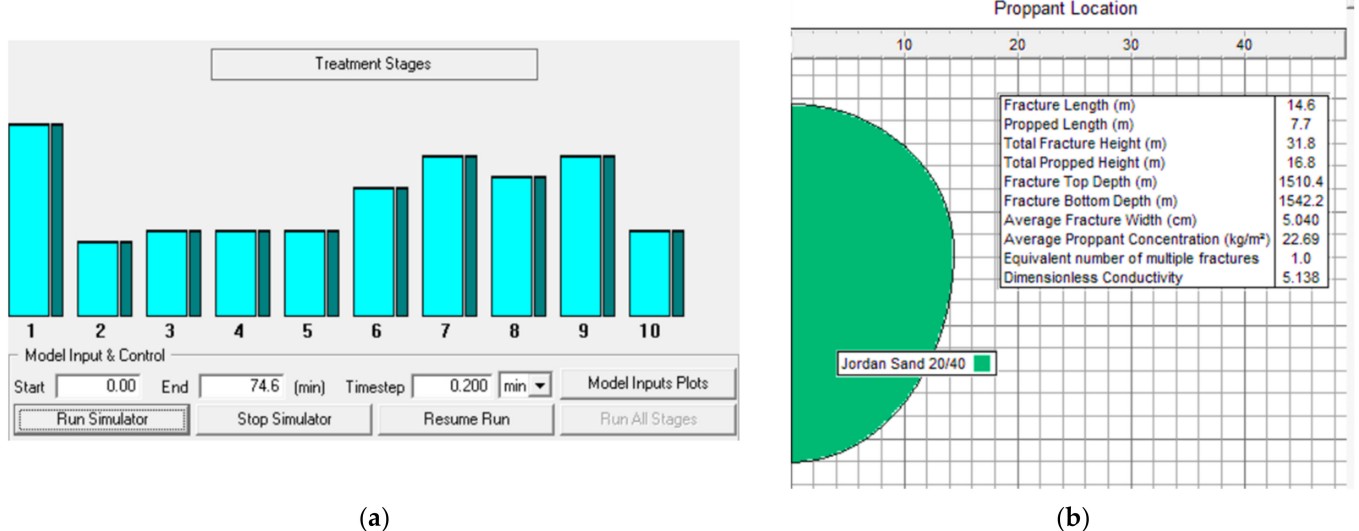

(**a**)                    (**b**)

**Figure 4.** (**a**) Fracpro program interface, treatment stages; (**b**) proppant distribution after the fracture closing. Case 1.

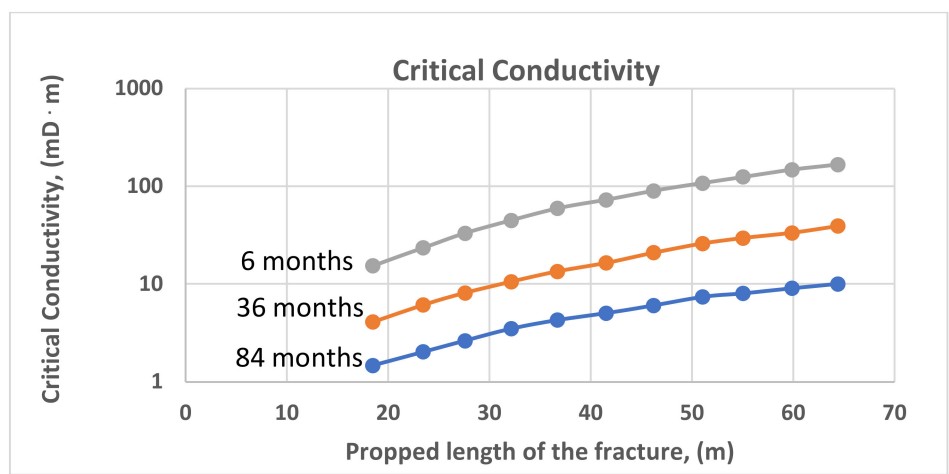

**Figure 5.** Selecting critical conductivity of the fracture function of the propped length of the fracture and number of production months.

The estimated production after fracturing (made in Fracpro production) can be observed in Table 5. The approximate estimated cost of the operation is 203,000 euros (some details involving cost elements are presented in Case 2). Following the production increase estimated in 2 years, 97,130 mSm³ at a natural gas price of 118 Euro/MWh (CEGH Central European Gas Hub last quarter of 2021, for an average gross calorific heat of

10,866 kWh/Smc) an amount of 124,538,920 euros is obtained. The value of the fracturing operation represents only 0.16% of the revenues obtained by increasing production. So, the benefits of the operation in terms of current gas prices are spectacular. There are small differences between the predicted production values and those recorded after the hydraulic fracturing operation, as demonstrated in Figure 7. It is also observed that after 20 months, the well stopped production.

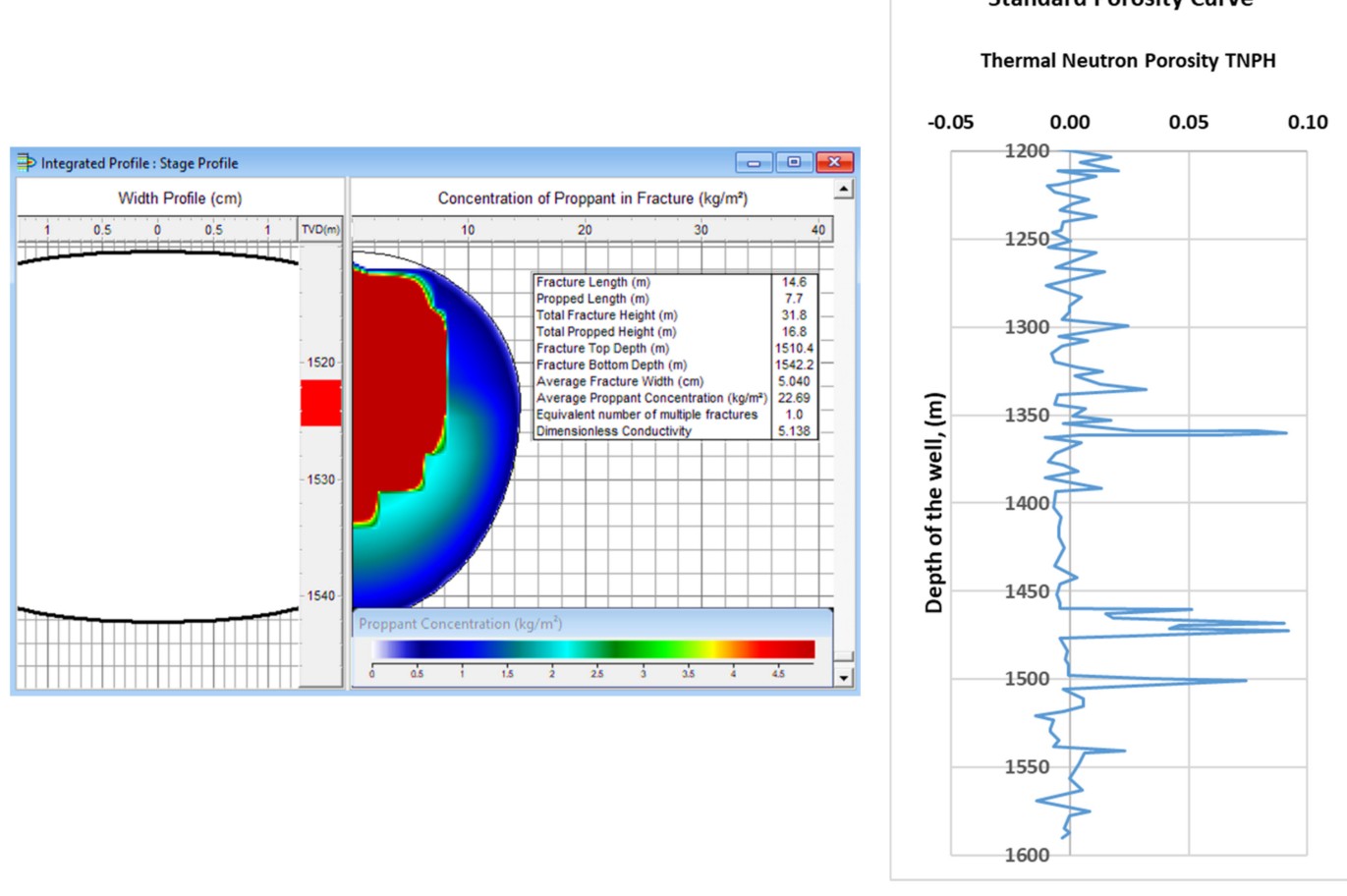

(**a**)                         (**b**)

**Figure 6.** *Cont.*

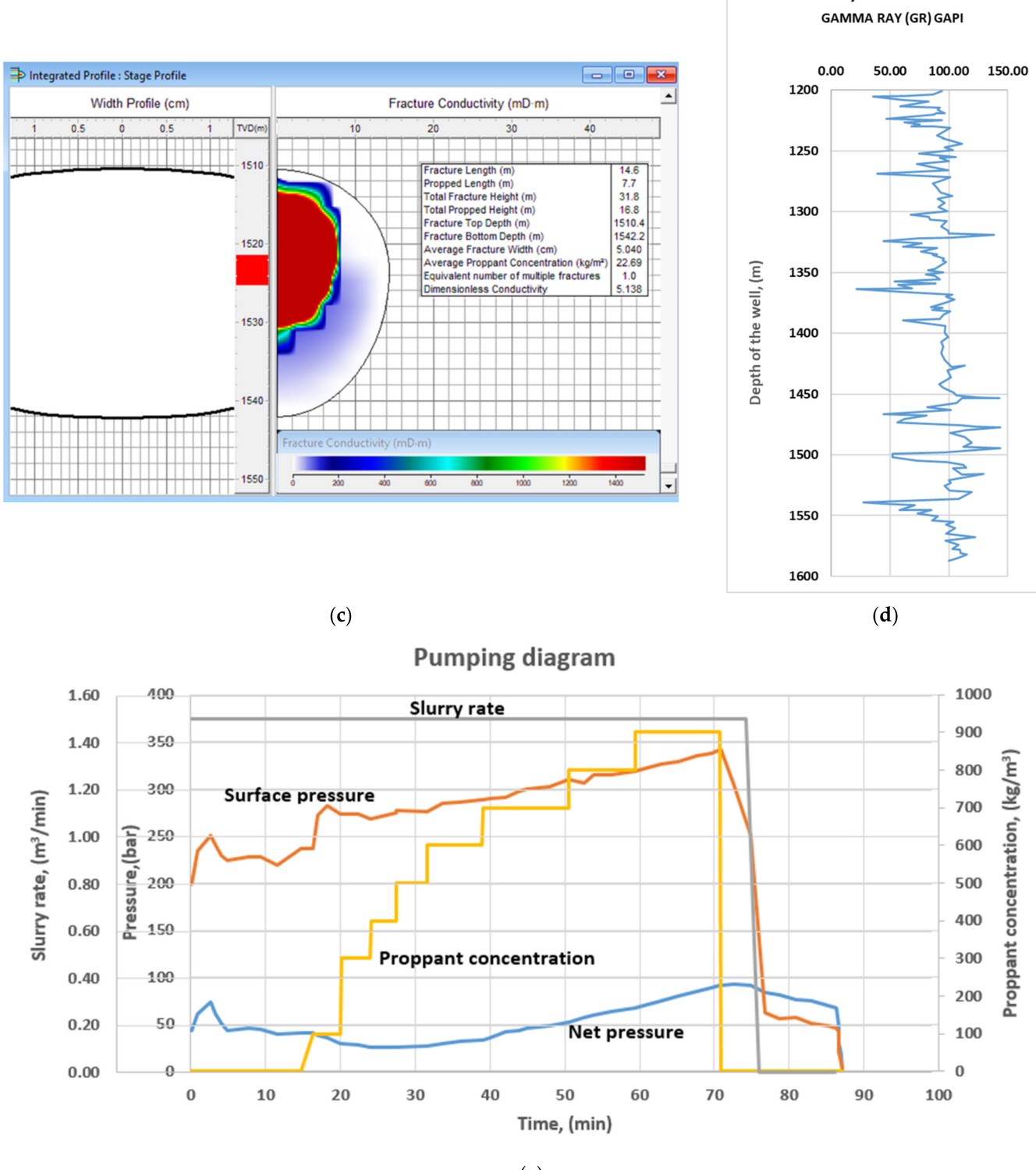

**Figure 6.** Examples of the analysis made in Fracpro for the Case 1: (**a**) proppant concentration and fracture width; (**b**) logging diagram for determining the permeability of the layers TNPH; (**c**) fracture conductivity and its width; (**d**) logging diagram for determining the permeability of the layers GAPI; (**e**) diagram of variation in pressure, proppant concentration and injection flow.

**Table 5.** Estimation of the production increase after hydraulic fracturing using Fracpro production analysis for natural gas well. Case 1.

| Time | Pressure in the Field | Estimated Gas Flow after Fracturing | Estimated Gas Flow without Fracturing | Additional Natural Gas Production | Time | Pressure in the Field | Estimated Gas Flow after Fracturing | Estimated Gas Flow without Fracturing | Additional Natural Gas Production |
|---|---|---|---|---|---|---|---|---|---|
| months | bar | $Sm^3$/day | $Sm^3$/day | $mSm^3$/month | months | bar | $Sm^3$/day | $Sm^3$/day | $mSm^3$/month |
| 1 | 120 | 43,010 | 25,300 | 531 | 13 | 81 | 9426 | 2893 | 4572 |
| 2 | 115 | 38,740 | 21,682 | 1043 | 14 | 80 | 8397 | 2398 | 4752 |
| 3 | 110 | 34,460 | 17,891 | 1540 | 15 | 79 | 7506 | 1988 | 4918 |
| 4 | 105 | 30,129 | 14,779 | 2000 | 16 | 78 | 6734 | 1648 | 5071 |
| 5 | 100 | 26,583 | 13,064 | 2406 | 17 | 77 | 6064 | 1365 | 5212 |
| 6 | 97 | 22,808 | 10,809 | 2766 | 18 | 76 | 5484 | 1131 | 5342 |
| 7 | 93 | 19,926 | 8948 | 3095 | 19 | 76 | 4982 | 939 | 5463 |
| 8 | 90 | 17,496 | 7409 | 3398 | 20 | 75 | 4546 | 778 | 5576 |
| 9 | 88 | 15,390 | 6137 | 3675 | 21 | 75 | 4169 | 644 | 5682 |
| 10 | 86 | 13,565 | 5084 | 3930 | 22 | 74 | 3842 | 535 | 5781 |
| 11 | 84 | 11,984 | 4213 | 4163 | 23 | 74 | 3558 | 443 | 5875 |
| 12 | 82 | 10,613 | 3491 | 4376 | 24 | 74 | 3313 | 368 | 5963 |
| | | | | Total | | | | | 97,130 |

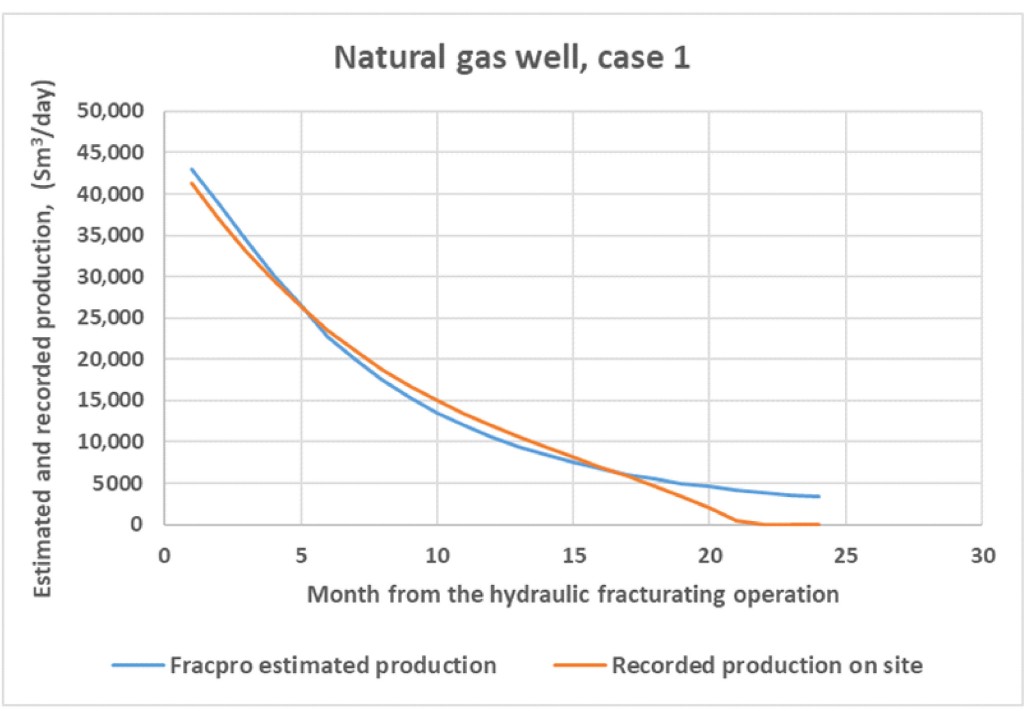

**Figure 7.** Comparison between the estimated natural gas production and the recorded production, for 24 months. Case 1.

### 3.2. Case 2. BRADU Well

The sketch of the surface equipment is shown in Figure 8a. The equipment includes (1,2,3) the main pumps for fracturing; (4) a hydration unit; (5) a filtering and heating unit; (6) a pump on the truck, (7) the main valve; (8) fracturing solution tanks; (9) a freshwater tank; (10) a filtered brine tank; (11) a sand truck; (12) a blender; (13) a data acquisition cabin; (14) a gas/oil well. The diagram of the down-the-hole equipment in the area of fractures is shown in Figure 8b.

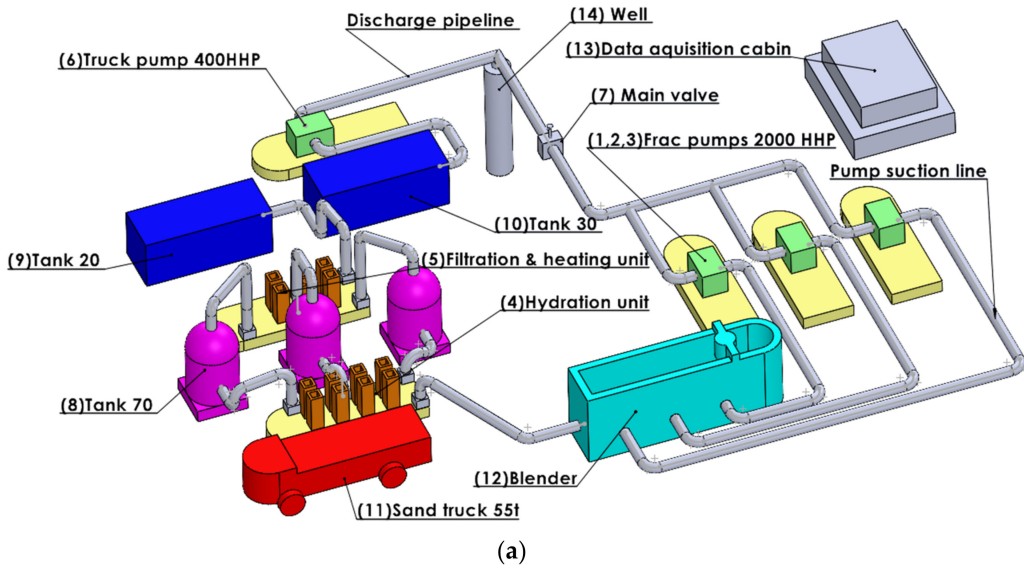

(**a**)

**Figure 8.** *Cont.*

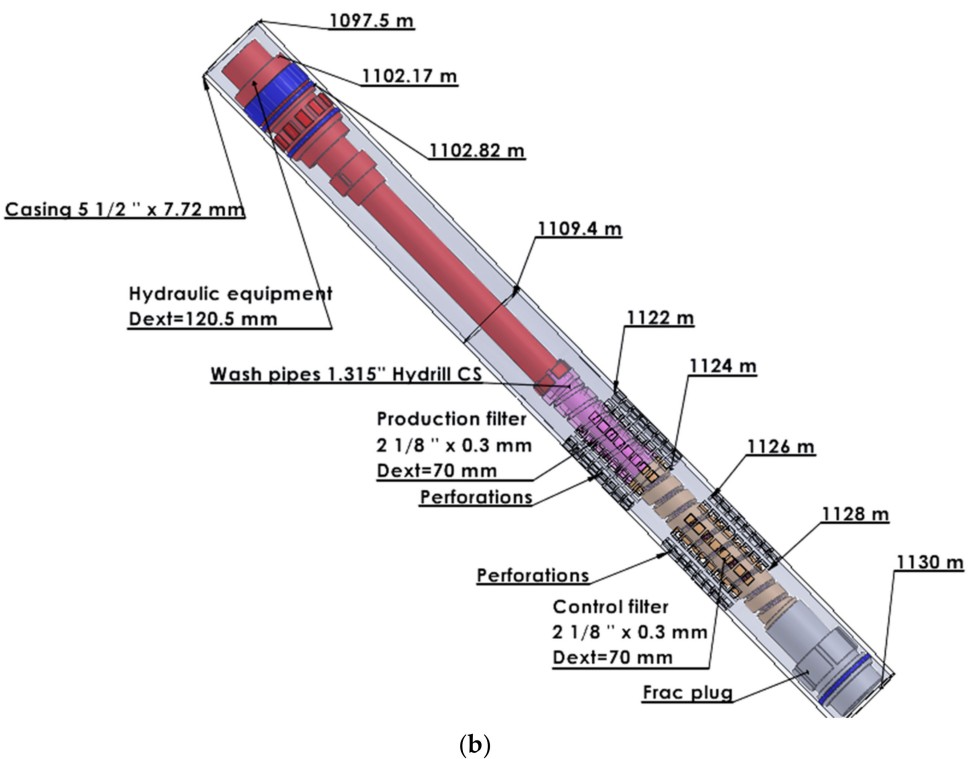

(**b**)

**Figure 8.** Details of the equipment used: (**a**) surface equipment layout for hydraulic fracturing; (**b**) schematic of the down-the-hole assembly for the operation of frac-pack.

The stimulation operation aims to increase the productivity index of the well, applying a sustained hydraulic stimulation treatment. The deposit is of the lower Meotian type b, weakly calcareous sands with marl inserts. The average deposit temperature is 42 °C, average porosity 22%, and permeability 85 mD. The well has crude oil production. The geometric elements of the well, casing, tubing, and the perforations are given in Figure 9.

The choice of fracturing fluid is made while taking into account the following criteria: the filtrate of the fluid must be small enough to allow the extension of the crack along the designed length; to have a low blockage.

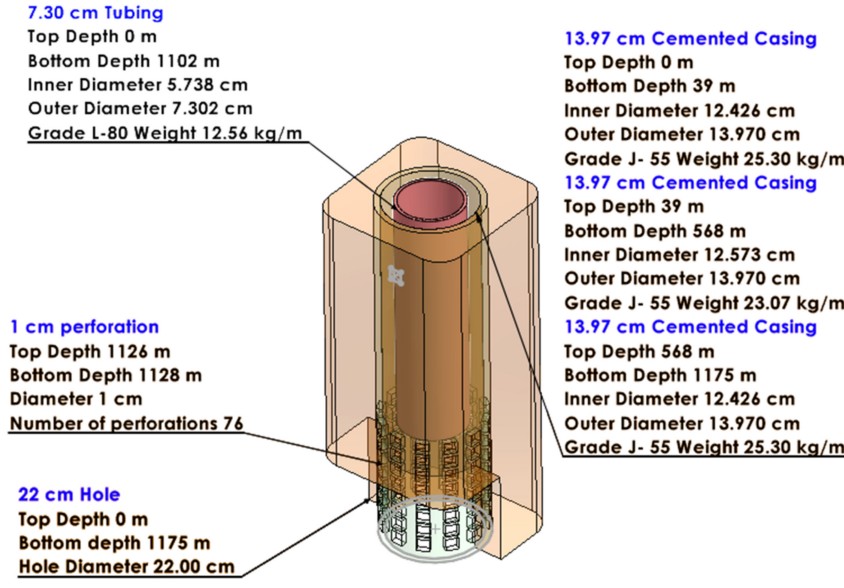

**Figure 9.** Geometric elements of the well, casing, tubing, and perforations. Case 2.

A linear PGHT30 CO2 30 fluid was chosen. The support material was chosen using the following criteria: its strength should be higher than the closing pressure of the crack; to have a high permeability at the crack closing pressure over 1350 mD · m. The chosen support material was Carbo Lite 16/20. The operating program was carried out according to the steps described in case no.1. The step-by-step flow variation test was performed before fracturing to determine both FEP and FCP, as shown in Table 6.

**Table 6.** Pumping program for step-by-step flow variation test (linear fluid). Case 2.

| Cycle | Stage | Injection Flow | Injection Time | Planned Volume | Total Planned Volume |
|-------|-------|----------------|----------------|----------------|----------------------|
|       |       | $m^3/min$ | s | $m^3$ | $m^3$ |
| Step up | Rate 1 | 0.20 | 60 | 0.20 | 0.20 |
|         | Rate 2 | 0.40 | 60 | 0.40 | 0.60 |
|         | Rate 3 | 0.60 | 60 | 0.60 | 1.20 |
|         | Rate 4 | 0.80 | 30 | 0.40 | 1.60 |
|         | Rate 5 | 1.20 | 30 | 0.60 | 2.20 |
|         | Rate 6 | 1.40 | 30 | 0.70 | 2.90 |
|         | Rate 7 | 1.60 | 30 | 0.80 | 3.70 |
|         | Rate 8 | 1.80 | 30 | 0.90 | 4.60 |
| Step down | Rate 9 | 1.60 | 15 | 0.40 | 5.00 |
|           | Rate 10 | 1.40 | 15 | 0.35 | 5.35 |
|           | Rate 11 | 1.00 | 15 | 0.25 | 5.60 |
|           | Rate 12 | 0.60 | 15 | 0.15 | 5.75 |

The operating fluid (filtered reservoir water $995 \frac{l}{m^3}$ and Brtentahib $5 \frac{l}{m^3}$) is injected into the formation, increasing the flow in steps, as shown in Table 6. Each flow step is maintained until the injection pressure stabilizes and these values are recorded. The bottom pressure graph is drawn as a function of the injection flow rate and the flow rate and fracture initiation pressure corresponding to the slope change point is determined. FEP was 146 bar and the value of FCP was 123 bar.

The mini-fracture analysis is based on the data in Table 7. After the fluid is pumped to the mini-frac, the fracture is expected to close and the pore pressure drop is recorded. After analyzing the decline data, the operation is redesigned with the new data. The characteristics of the rock zones traversed by the oil well in the fracturing zone are indicated in Table 8. The initiation and extension of the fracture are carried out according to the treatment program indicated in Table 9 and Figure 10a. The fracture closing time is 69.2 min. The proppant location in the fracture is shown in Figure 10b.

**Table 7.** Mini-frac pumping program (linear fluid). Case 2.

| Stage | Fluid | Planned Volume | Proppant Concentration | Injection Flow | Proppant Type |
|-------|-------|----------------|------------------------|----------------|---------------|
|       |       | $m^3$ | $\frac{kg}{m^3}$ | $m^3/min$ | - |
| Pad | Reticulated gel | 5 | 0 | 1.80 | - |
| Sand plug | Reticulated gel | 5 | 120 | 1.80 | 16/20 CarboLite |
| Pad | Reticulated gel | 5 | 0 | 1.80 | - |
| Flush | Linear gel | 4 | 0 | 1.80 | - |
| Shut-in | - | 0 | 0 | 0 | - |

**Table 8.** Characteristics of the layers traversed by crude oil well in the fracture zone. Case 2.

| Depth TVD | Depth MD | Layer Thickness | Rock Type | Pore Fluid Permeability | Leak-Off Coefficient | Stress | Young's Modulus | Poisson's Ration | Fracture Toughness |
|-----------|----------|-----------------|-----------|-------------------------|----------------------|--------|-----------------|------------------|--------------------|
| m | m | m | - | mD | $m \cdot min^{1/2}$ | bar | MPa | - | $MPa \cdot m/s$ |
| 840 | 840 | 2 | Sandstone | 60 | 1393 | 133.2 | 13,800 | 0.20 | 0.11 |

**Table 8.** *Cont.*

| Depth TVD | Depth MD | Layer Thickness | Rock Type | Pore Fluid Permeability | Leak-Off Coefficient | Stress | Young's Modulus | Poisson's Ration | Fracture Toughness |
|---|---|---|---|---|---|---|---|---|---|
| 842 | 842 | 3 | Shale | 0 | 0 | 152.6 | 17,200 | 0.25 | 0.22 |
| 845 | 845 | 7 | Sandstone w/Water | 59.8 | 1393 | 134.4 | 13,800 | 0.20 | 0.11 |
| 852 | 852 | 23 | Shale | 0 | 0 | 156.3 | 17,200 | 0.25 | 0.22 |
| 875 | 875 | 245 | Sandstone | 85 | 1428 | 157.9 | 13,800 | 0.20 | 0.11 |
| 1120 | 1120 | 2 | Shale | 0 | 0 | 202.9 | 17,200 | 0.25 | 0.22 |
| 1122 | 1122 | 3 | Sandstone | 85 | 1428 | 177.9 | 13,800 | 0.20 | 0.11 |
| 1125 | 1125 | 1 | Shale | 0 | 0 | 203.7 | 17,200 | 0.25 | 0.22 |
| 1126 | 1126 | 1 | Sandstone | 85 | 1457 | 178.4 | 13,800 | 0.20 | 0.11 |
| 1127 | 1127 | 3 | Shale | 0 | 0 | 204.2 | 17,200 | 0.25 | 0.22 |
| 1130 | 1130 | 45 | Sandstone | 85.5 | 1428 | 182.5 | 13,800 | 0.20 | 0.11 |
| 1175 | 1175 | 0 | Sandstone | 85 | 1428 | 186.1 | 13,800 | 0.20 | 0.11 |

**Table 9.** The hydraulic fracturing treatment from the Fracpro simulator. Case 2.

| Stage Type | Flow Rate | Proppant Concentration | Clean Volume | Stage Length | Cumul Time | Fluid Type | Proppant Type |
|---|---|---|---|---|---|---|---|
| | m³/min | g/l | m³ | min | min:sec | | |
| Main frac pad | 1.80 | 0 | 25.00 | 13.89 | 13:53 | PGHT30 CO2 30 | |
| Main frac slurry | 1.80 | 120 | 5.00 | 2.90 | 16:46 | PGHT30 CO2 30 | CarboLite16/20 |
| Main frac slurry | 1.80 | 240 | 6.00 | 3.63 | 20:25 | PGHT30 CO2 30 | CarboLite16/20 |
| Main frac slurry | 1.80 | 360 | 7.00 | 4.40 | 24:49 | PGHT30 CO2 30 | CarboLite16/20 |
| Main frac slurry | 1.80 | 480 | 8.00 | 5.23 | 30:03 | PGHT30 CO2 30 | CarboLite16/20 |
| Main frac slurry | 1.80 | 600 | 10.00 | 6.78 | 36:50 | PGHT30 CO2 30 | CarboLite16/20 |
| Main frac slurry | 1.80 | 720 | 10.00 | 7.03 | 43:51 | PGHT30 CO2 30 | CarboLite16/20 |
| Main frac slurry | 1.80 | 840 | 12.00 | 8.73 | 52:35 | PGHT30 CO2 30 | CarboLite16/20 |
| Main frac slurry | 1.80 | 960 | 14.00 | 10.53 | 63:07 | PGHT30 CO2 30 | CarboLite16/20 |
| Shut-in | 1.80 | 0 | 3.00 | 1.67 | 64:47 | PGHT30 CO2 30 | |

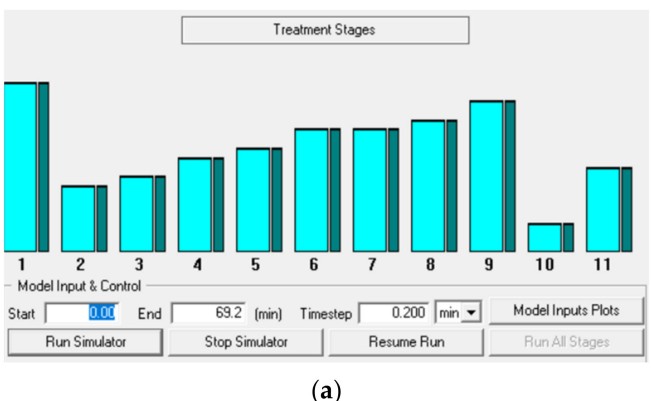

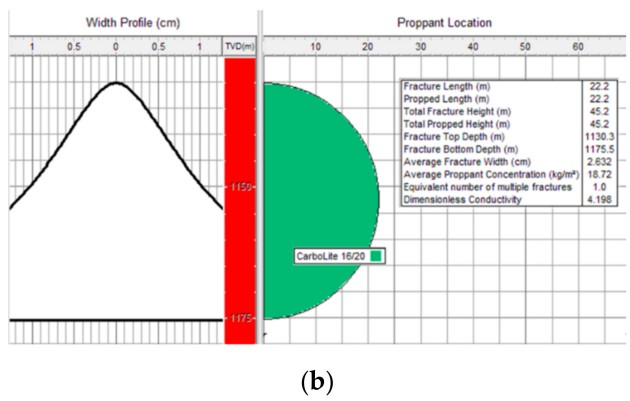

(**a**)  (**b**)

**Figure 10.** (**a**) Fracpro program interface, treatment stages; (**b**) proppant distribution after the fracture closing. Case 2.

The data used in the Fracpro simulator were selected as in the Case 1. The use of the Fracpro program makes it possible to anticipate the size of the fracture, the concentration of proppant in the fracture, and its conductivity. Fracture parameters between 1222 m and 1226 m (Figure 11a,c and application report) are as follows: half from the fracture length, 22 m; fracture height at the end of the operation, 45.2 m; half the sustained width of the fracture, 22 m; sustained fracture height, 45.2 m; the upper limit of the created fracture, 1130.3 m; the lower limit of the created fracture, 1175.5 m; the depth of the fracture sustained at the top, 1130.3; the depth of the fracture sustained at the base, 1175.5 m; average surface pressure, 145 bar; average fracture conductivity, 7897.9 mD · m; dimensionless fracture conductivity, 4.17; maximum fracture width, 43.14 cm; average fracture width, 2.63 cm; average proppant concentration in the fracture, 18.72 $\frac{kg}{m^2}$; average hydraulic power, 153 kW; total proppant pumped, 30,091 kg; total proppant in fracture, 29,572 kg. The logging diagrams of the well are shown in Figure 11b,d. Regarding the pressure variation, as shown in Figure 11e, there is an increase in this at the beginning of the fracture, after which the pressure decreases continuously until the middle of the operation. Following the extension of the fracture, the pressure increases again to values close to those at the beginning of the cycle. The approximate estimated cost of the operation is 174,340 euros, as demonstrated in Table 10.

**Table 10.** Estimation of costs for hydraulic fracturing for case 2.

| Description | Price [Euro] |
| --- | --- |
| Estimated cost for operating fluid (60 m$^3$) | 1478 |
| Estimated cost for acid solution (2 m$^3$) | 2216 |
| Estimated cost for fracturing company services | 128,539 |
| Estimated cost for Frac-Pack equipment | 36,937 |
| Estimated cost of frac-pack operation design | 1478 |
| Estimated cost of technical assistance HQ stimulation | 3694 |
| Estimated total cost | 174,340 |

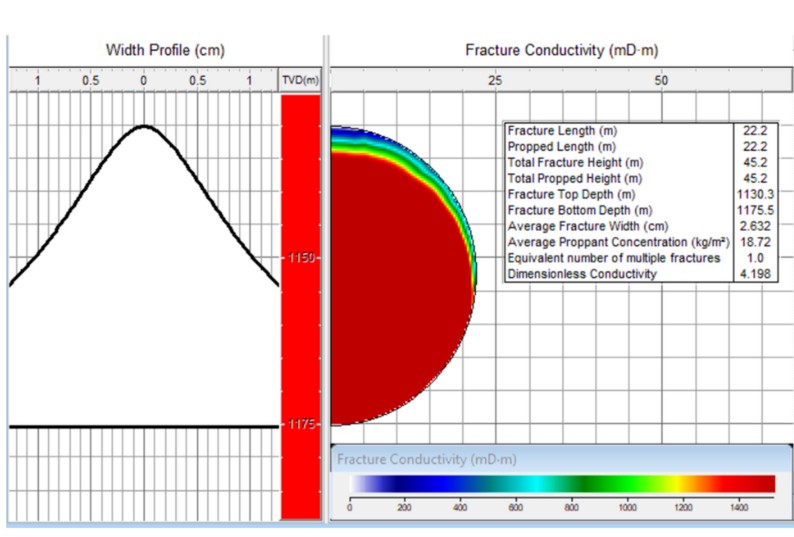

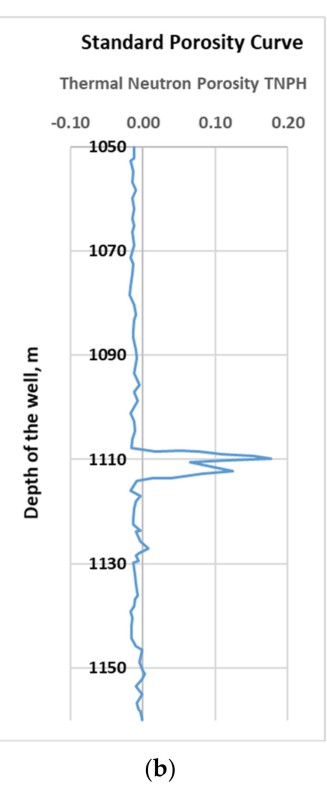

(a)

(b)

**Figure 11.** *Cont.*

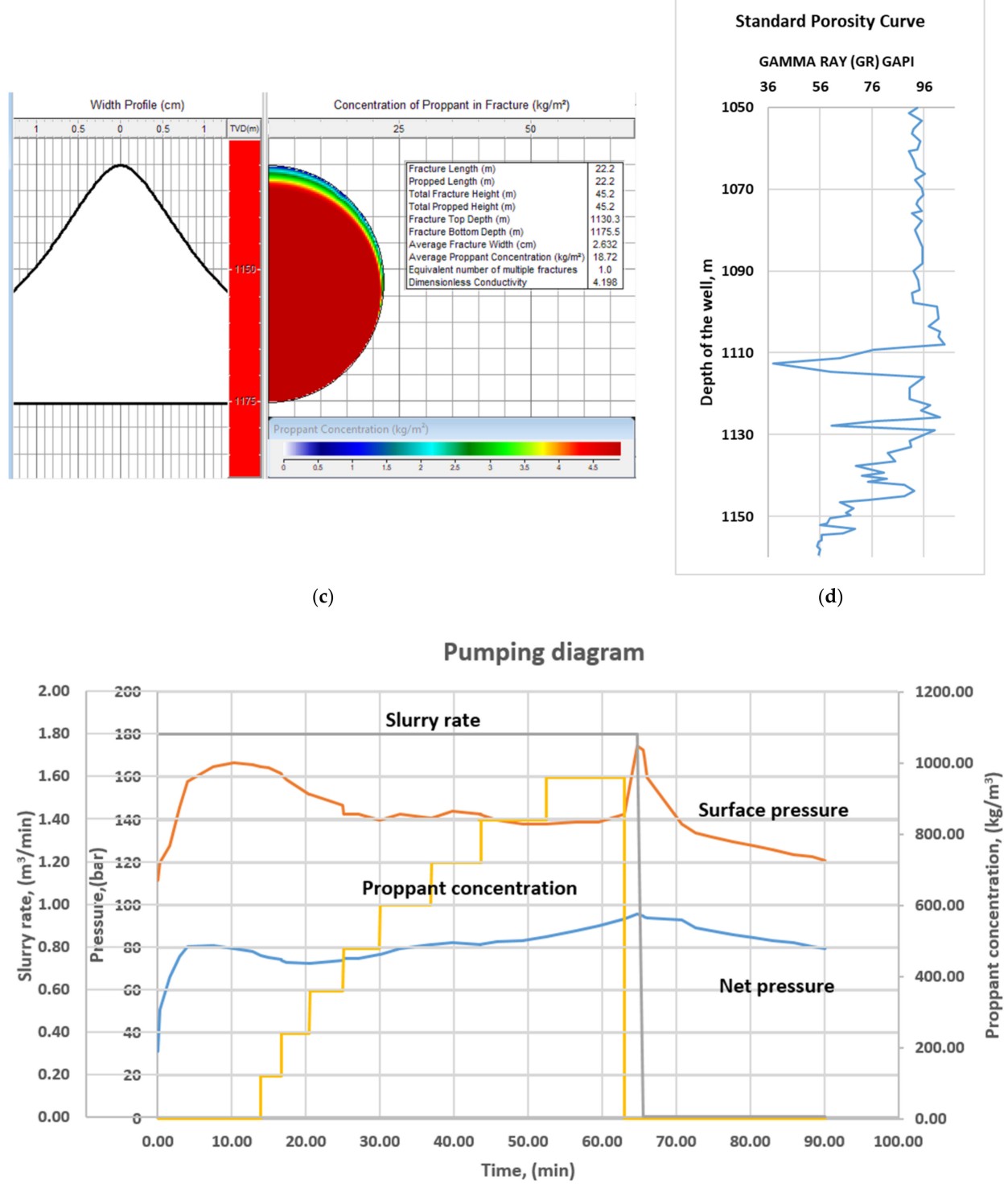

**Figure 11.** Examples of the analysis made in Fracpro for Case 2: (**a**) proppant concentration and fracture width; (**b**) logging diagram for determining the permeability of the layers TNPH; (**c**) fracture conductivity and its width; (**d**) logging diagram for determining the permeability of the layers GAPI; (**e**) diagram of variation in pressure, proppant concentration and injection flow.

The cost includes several categories of the fracturing operation costs, such as operating fluid; acid solution; fracturing company services; frac-pack equipment; frac-pack operation design; and technical assistance HQ stimulation.

### 3.3. Case 3. MOINESTI Well

For the third case analyzed, we have a Kliwa 2 and Kliwa 3 deposit. The average deposit temperature is 32 °C, average porosity 15% and permeability 29–60 mD. The well has a crude oil production. The geometric elements of the well, casing, tubing, and perforations are given in Figure 12.

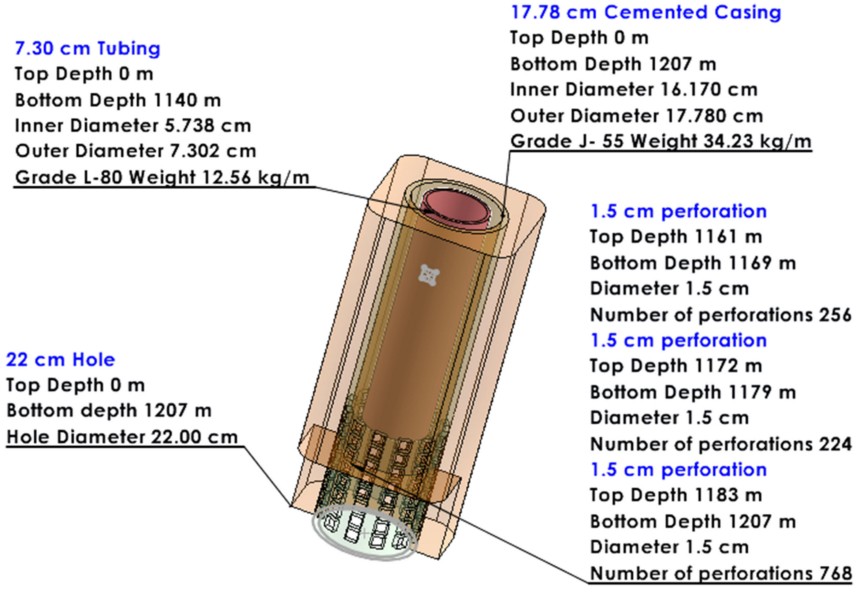

**Figure 12.** Geometric elements of the well, casing, tubing, and perforations for Case 3.

The fracturing fluid and the support material were chosen as in the previous cases. The fluid HG 40 was chosen for fracturing and the selected proppant was Jordan Sand 12/20 in the main frac slurry phases and Carbo Lite 12/18 in the phase main frac flush. The step-by-step flow variation test is performed before cracking to determine both FEP and FCP. The operating fluid filtered saltwater 995 $\frac{l}{m^3}$ and Brtentahib 5 $\frac{l}{m^3}$ is injected into the formation, increasing the flow in steps, as shown in Table 11. Each flow step is maintained until the injection pressure is stabilized and these values are recorded. The bottom pressure graph is drawn as a function of the injection flow rate and the flow rate and crack initiation pressure corresponding to the slope change point is determined. FEP was 225 bar and the value of FCP was 180 bar.

The mini-frac analysis is performed based on the data in Table 12. After pumping the fluid to the mini-frac, the closure of the fracture is expected and the pressure drop is recorded. After analyzing the pressure data, the operation is redesigned with the new data.

The characteristics of the rock zones traversed by the crude oil well in the fracturing zone are indicated in Table 13. The treatment designed program is presented in Table 14 and Figure 13a. The distribution of the proppant in the fracture is in Figure 13b. The data used in the Fracpro simulator were selected as in the previous cases. The use of the Fracpro program makes it possible to anticipate the size of the fracture, the concentration of proppant in the fracture, and fracture conductivity. Fracture parameters at the intervals 1161–1169 m; 1172–1179 m; and 1183–1207 m (Figure 14a,c and application report) are as follows: half from the fracture length, 6.9 m; 6.3 m; 13 m; fracture height at the end of the operation, 13.9 m; 12.7 m; 26 m; the sustained width of the fracture, 6.9 m; 6.3 m; 26 m; sustained fracture height, 13.9 m; 12.7 m; 26 m; the upper limit of the created fracture, 1158.1 m; 1169.7 m; 1181 m; the lower limit of the created fracture, 1171.9 m; 1182.3 m; 1207 m; the depth of the fracture sustained at the top, 1158.1 m; 1169.7 m; 1181 m; the depth of the fracture sustained at the base, 1171.9 m; 1182.3 m; 1207 m; average surface pressure, 195 bar; average fracture conductivity, 7753.8; 9180.5; 18,447.5 mD·m; fracture closing time, 56.5 min; maximum fracture width, 9.16 cm; 3.00 cm; 9.32 cm; average fracture

width, 0.77 cm; 2.01 cm; 6.27 cm; average proppant concentration in the fracture, 6.57; 17.06; 48.91 $\frac{kg}{m^2}$; average hydraulic power, 2533 kW; total proppant pumped, 34,400 kg; total proppant in fracture, 26,000 kg. The logging diagram of the well is shown in Figure 14b,d.

**Table 11.** Pumping program for step-by-step flow variation test for Case 3.

| Cycle | Stage | Injection Flow | Injection Time | Planned Volume | Total Planned Volume |
|---|---|---|---|---|---|
| | - | $m^3/min$ | s | $m^3$ | $m^3$ |
| Step up | Rate 1 | 0.20 | 60 | 0.20 | 0.20 |
| | Rate 2 | 0.40 | 60 | 0.40 | 0.60 |
| | Rate 3 | 0.60 | 60 | 0.60 | 1.20 |
| | Rate 4 | 0.80 | 30 | 0.40 | 1.60 |
| | Rate 5 | 1.20 | 30 | 0.60 | 2.20 |
| | Rate 6 | 1.40 | 30 | 0.70 | 2.90 |
| | Rate 7 | 1.60 | 30 | 0.80 | 3.70 |
| | Rate 8 | 1.80 | 30 | 0.90 | 4.60 |
| Step down | Rate 9 | 1.60 | 15 | 0.40 | 5.00 |
| | Rate 10 | 1.40 | 15 | 0.35 | 5.35 |
| | Rate 11 | 1.00 | 15 | 0.25 | 5.60 |
| | Rate 12 | 0.60 | 15 | 0.15 | 5.75 |

**Table 12.** Mini-frac pumping program (linear fluid) for Case 3.

| Stage | Fluid | Planned Volume | Propant Concentration | Injection Flow | Proppant Type |
|---|---|---|---|---|---|
| - | - | $m^3$ | $\frac{kg}{m^3}$ | $m^3/min$ | - |
| Pad | Reticulated gel | 5 | 0 | 1.80 | - |
| Sand plug | Reticulated gel | 5 | 120 | 1.80 | 16/20 CarboLite |
| Pad | Reticulated gel | 5 | 0 | 1.80 | - |
| Flush | Linear gel | 4 | 0 | 1.80 | - |
| Shut-in | - | 0 | 0 | 0 | - |

**Table 13.** Characteristics of the layers traversed by crude oil well in the fracture zone. Case 3.

| Depth TVD | Depth MD | Layer Thickness | Rock Type | Pore Fluid Permeability | Leak-off Coefficient | Stress | Young's Modulus | Poisson's Ration | Fracture Toughness |
|---|---|---|---|---|---|---|---|---|---|
| m | m | m | - | mD | $m \cdot min^{1/2}$ | bar | MPa | - | $MPa \cdot m/s$ |
| 845 | 845 | 15 | Sandstone w/Water | 28 | 1417 | 134.4 | 13,800 | 0.20 | 0.11 |
| 859 | 859 | 16 | Shale | 0 | 0 | 206.3 | 17,200 | 0.25 | 0.22 |
| 875 | 875 | 6 | Sandstone | 60 | 1537 | 139.0 | 13,800 | 0.20 | 0.11 |
| 881 | 881 | 274 | Shale | 0 | 0 | 204.2 | 17,200 | 0.25 | 0.22 |
| 1155 | 1155 | 14 | Sandstone | 55 | 1525 | 184.0 | 13,800 | 0.20 | 0.11 |
| 1169 | 1169 | 2 | Shale | 0 | 0 | 211.7 | 17,200 | 0.25 | 0.22 |
| 1171 | 1171 | 8 | Sandstone | 28 | 1417 | 186.1 | 13,800 | 0.20 | 0.11 |
| 1179 | 1179 | 4 | Shale | 0 | 0 | 213.7 | 17,200 | 0.25 | 0.22 |
| 1183 | 1183 | 24 | Sandstone | 29 | 1423 | 189.2 | 13,800 | 0.20 | 0.11 |
| 1207 | 1207 | 0 | Sandstone | 32 | 1441 | 191.1 | 13,800 | 0.20 | 0.11 |

**Table 14.** The hydraulic fracturing treatment from the Fracpro simulator. Case 3.

| Stage Type | Flow Rate | Proppant Concentration | Clean Volume | Stage Length | Cumul Time | Fluid Type | Proppant Type |
|---|---|---|---|---|---|---|---|
| | m³/min | g/l | m³ | min | min:sec | | |
| Main frac pad | 3.50 | 0 | 20.0 | 5.71 | 5:42 | HG_40 | |
| Main frac slurry | 3.50 | 100 | 15.0 | 4.42 | 10:08 | HG_40 | Jordan Sand 10/20 |
| Main frac slurry | 3.50 | 0 | 20.0 | 5.71 | 15:51 | HG_40 | Jordan Sand 10/20 |
| Main frac slurry | 3.50 | 200 | 10.0 | 3.04 | 18:53 | HG_40 | Jordan Sand 10/20 |
| Main frac slurry | 3.50 | 200 | 12.0 | 3.65 | 22:32 | HG_40 | Jordan Sand 10/20 |
| Main frac slurry | 3.50 | 400 | 14.0 | 4.52 | 27:04 | HG_40 | Jordan Sand 10/20 |
| Main frac slurry | 3.50 | 500 | 18.0 | 5.98 | 33:02 | HG_40 | Jordan Sand 10/20 |
| Main frac slurry | 3.50 | 600 | 20.0 | 6.83 | 39:52 | HG_40 | Jordan Sand 10/20 |
| Main frac slurry | 3.50 | 700 | 15.0 | 5.38 | 45:15 | HG_40 | CarboLite 12/18 |
| Shut-in | 3.50 | 0 | 3.0 | 0.86 | 46:06 | HG_40 | |

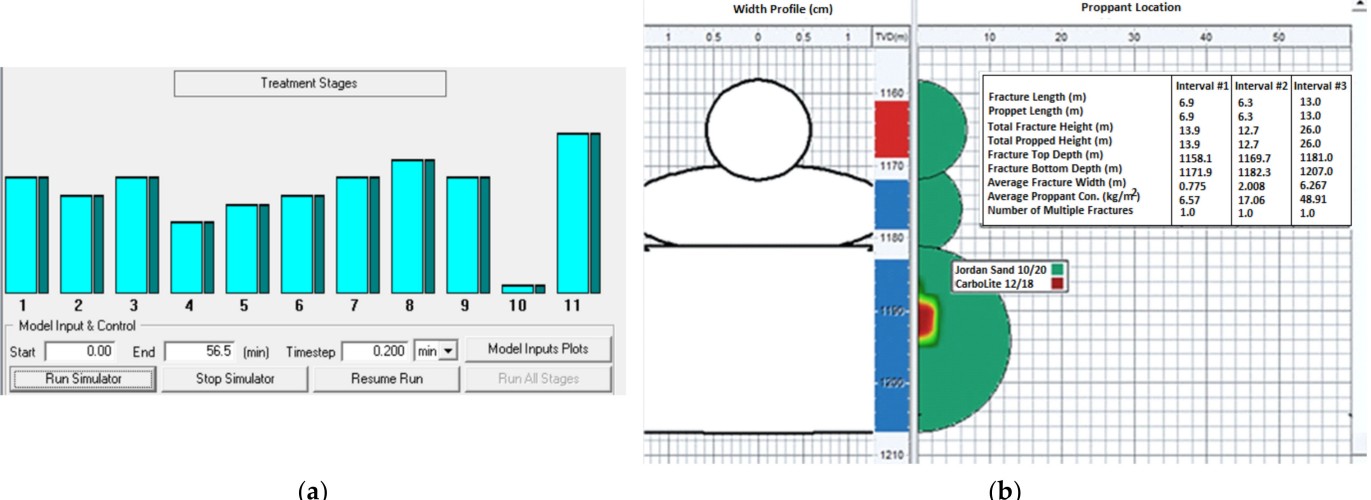

(**a**)  (**b**)

**Figure 13.** (**a**) Fracpro program interface, treatment stages; (**b**) proppant distribution after the fracture closing. Case 3.

The approximate estimated cost of the operation is 167,000 euros. The benefits obtained by carrying out the fracturing program are appreciated with the help of the Fracpro program production module. The well undergoing the operation is a crude oil well. Table 15 shows the economic calculations for this oil well. It is found that, in two years, the price of the fracturing operation is recovered and an amount equal to the cost of the fracturing is earned. There are differences between the predicted oil production values and those recorded after the hydraulic friction operation, as shown in Figure 15. The real production values are 10–27% lower. Under these conditions, the economic benefits are no longer as spectacular as in the case of 1. However, the costs are covered and the well continues to produce.

**Table 15.** Estimation of the oil production increase after hydraulic fracturing (Fracpro production analysis) for an oil well. Case 3.

| Time | Pressure in the Field | Estimated Oil Flow before Fracturing | Estimated Oil Flow after Fracturing | Additional Oil Production | Saltwater, 25% | Total Oil |
|---|---|---|---|---|---|---|
| months | bar | m³/day | m³/day | m³/month | m³ | m³ |
| 1 | 25.00 | 1.90 | 3.50 | 48.00 | 12.00 | 36.00 |
| 2 | 24.55 | 1.88 | 3.43 | 46.52 | 11.63 | 34.89 |
| 3 | 24.12 | 1.86 | 3.36 | 45.06 | 11.27 | 33.80 |
| 4 | 23.69 | 1.84 | 3.29 | 43.65 | 10.91 | 32.74 |
| 5 | 23.27 | 1.82 | 3.23 | 42.27 | 10.57 | 31.70 |
| 6 | 22.86 | 1.80 | 3.16 | 40.91 | 10.23 | 30.68 |
| 7 | 22.46 | 1.78 | 3.10 | 39.59 | 9.90 | 29.69 |
| 8 | 22.06 | 1.76 | 3.04 | 38.30 | 9.58 | 28.73 |
| 9 | 21.67 | 1.74 | 2.98 | 37.05 | 9.26 | 27.79 |
| 10 | 21.29 | 1.72 | 2.92 | 35.81 | 8.95 | 26.86 |
| 11 | 20.92 | 1.70 | 2.86 | 34.61 | 8.65 | 25.96 |
| 12 | 20.55 | 1.68 | 2.80 | 33.44 | 8.36 | 25.08 |
| 13 | 20.19 | 1.67 | 2.74 | 32.29 | 8.07 | 24.22 |
| 14 | 19.84 | 1.65 | 2.69 | 31.18 | 7.79 | 23.38 |
| 15 | 9.50 | 1.63 | 2.63 | 30.09 | 7.52 | 22.57 |
| 16 | 19.16 | 1.61 | 2.58 | 29.03 | 7.26 | 21.77 |
| 17 | 18.83 | 1.59 | 2.53 | 27.98 | 6.99 | 20.98 |
| 18 | 8.50 | 1.58 | 2.48 | 26.97 | 6.74 | 20.23 |
| 19 | 18.18 | 1.56 | 2.43 | 25.97 | 6.49 | 19.48 |
| 20 | 17.87 | 1.54 | 2.38 | 25.00 | 6.25 | 18.75 |
| 21 | 7.56 | 1.53 | 2.33 | 24.06 | 6.01 | 18.04 |
| 22 | 17.26 | 1.51 | 2.28 | 23.13 | 5.78 | 17.35 |
| 23 | 16.97 | 1.49 | 2.23 | 22.24 | 5.56 | 16.68 |
| 24 | 6.68 | 1.48 | 2.19 | 21.36 | 5.34 | 16.02 |
| Oil price per baril $/bbl February 2022 | 96.43 | | | | Total, m³ | 603.38 |
| | | | | | Total, bbl | 3794.8 |
| Fracturing Cost | Euro/$ rate = 1.13 February, 2022 | 167,000 Euros | 188,719$ | | Total Revenue, $ | 365,939 |

## 4. Conclusions

Hydraulic fracturing well treatment works, used in order to increase production, are not very numerous in Romania and those working in this field have gained some of the necessary experience through the tests carried out over the last decade. This article gives an overview of how fracturing technologies have been developed for three case studies. The main ideas were explained at the beginning of the third section.

Detailed knowledge of the structure of rock layers is mandatory for the application of the method. The simulator used in the design of the technology (Fracpro) requires a large quantity of measured or known data from the archives of drilling companies and Romanian national geologic data. Figures 1 and 2, together with the considerations at the beginning of point 3, present this aspect. These data are the support for the simulator used, as demonstrated in Tables 3, 8 and 13. In addition, we indicated the step-by-step flow variation tests realized before the hydraulic fracturing, as shown in Tables 1, 6 and 11. They are useful to establish the expansion and closure pressure into the fracture, with values indicated for each case.

The fracturing technology established after many attempts has some stages that are shown in Tables 4, 9 and 14, and Figures 4a, 10a and 13a. Their realization combines the following elements: a fracturing fluid, a proppant, the holding times for the fracturing phases, the required flows, the number of perforations, and their size. The simulators are essential in anticipating the shape and the characteristics of the fracture and in further

anticipating the production results. It is noted that this article did not present the known theoretical models from the presented bibliography, elements that are included in the computer products. These models are quite complicated, require many parameters and are sensitive to value ranges. The progress made in the development of theoretical support has been extensively presented in the bibliographic analysis. Many data can be obtained with the help of the simulator, some of which have been exemplified by Figure 4b, Figure 6a,c, Figure 10b, Figure 11a,c, Figures 13b and 14a,c. We are looking for good penetration of the proppant in the fracture and good conductivity, above 1350 mD · m. However, in many cases, these objectives can only be achieved with high pressures and flows and large quantities of proppant; therefore, the operation becomes too expensive or difficult to achieve from technical point of view. The diagrams in Figures 6e, 11e and 14e illustrate the evolution of some important parameters. In any case, the theoretical support implemented in the simulators provides a fairly quick answer, regarding the shape and parameters of the fracture. The team did not perform a laboratory analysis and the elements described were actually applied in production locations in Romania. The technology testing and last-minute corrections were carried out with the mini-frac program described in the analyzed cases, as shown in Tables 2, 7 and 12.

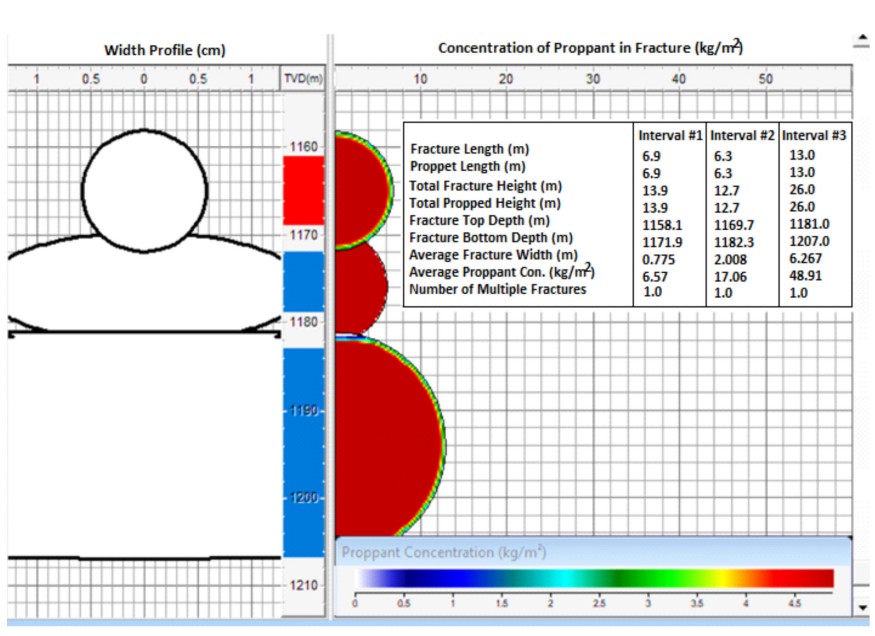

(a)

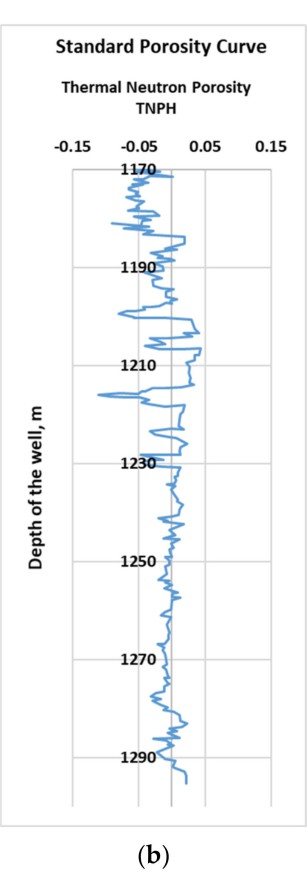

(b)

**Figure 14.** *Cont.*

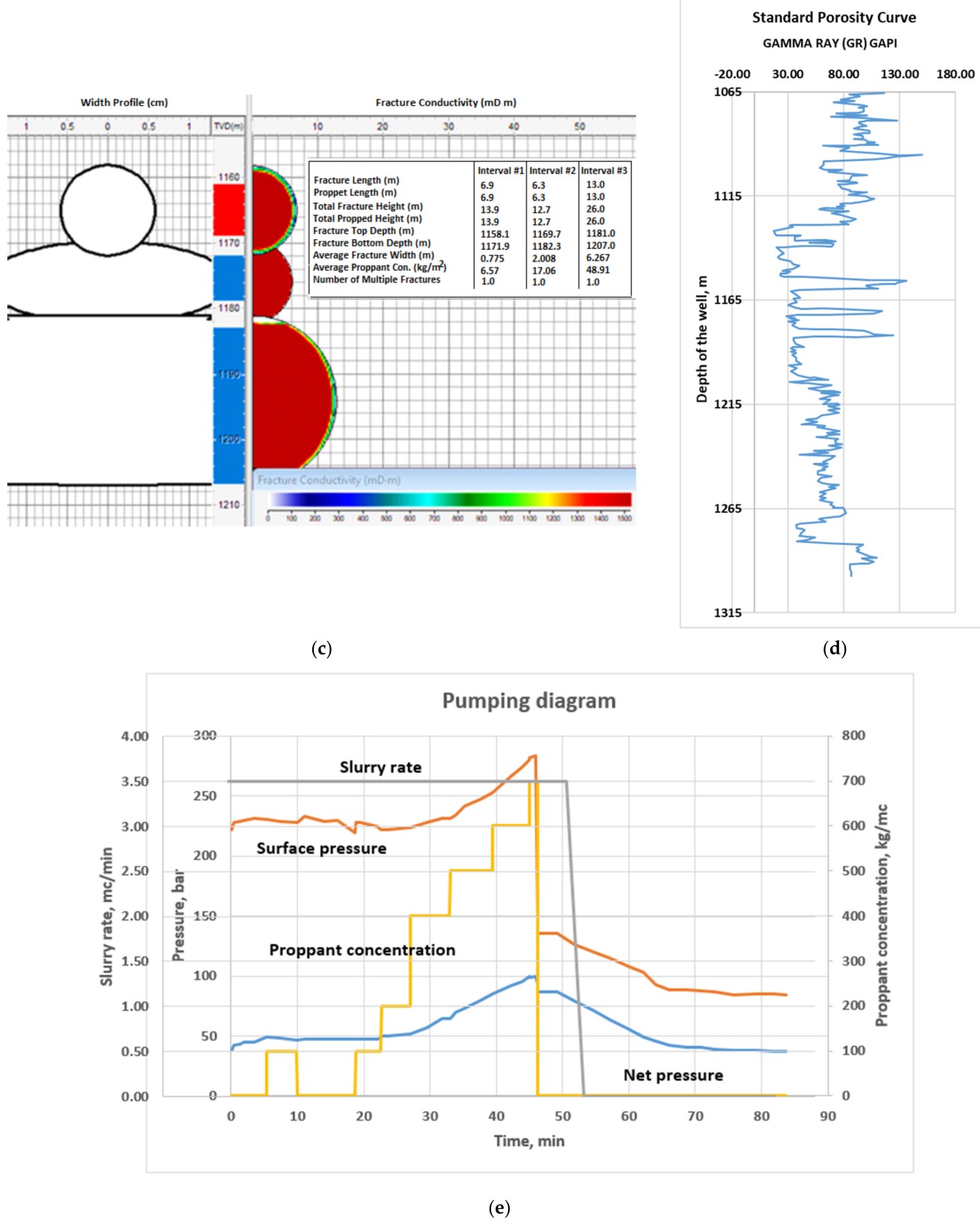

**Figure 14.** Examples of the analyses made in Fracpro for Case 3: (**a**) proppant concentration and fracture width; (**b**) logging diagram for determining the permeability of the layers TNPH; (**c**) fracture conductivity and its width; (**d**) logging diagram for determining the permeability of the layers GAPI; (**e**) diagram of variation in pressure, proppant concentration and injection flow.

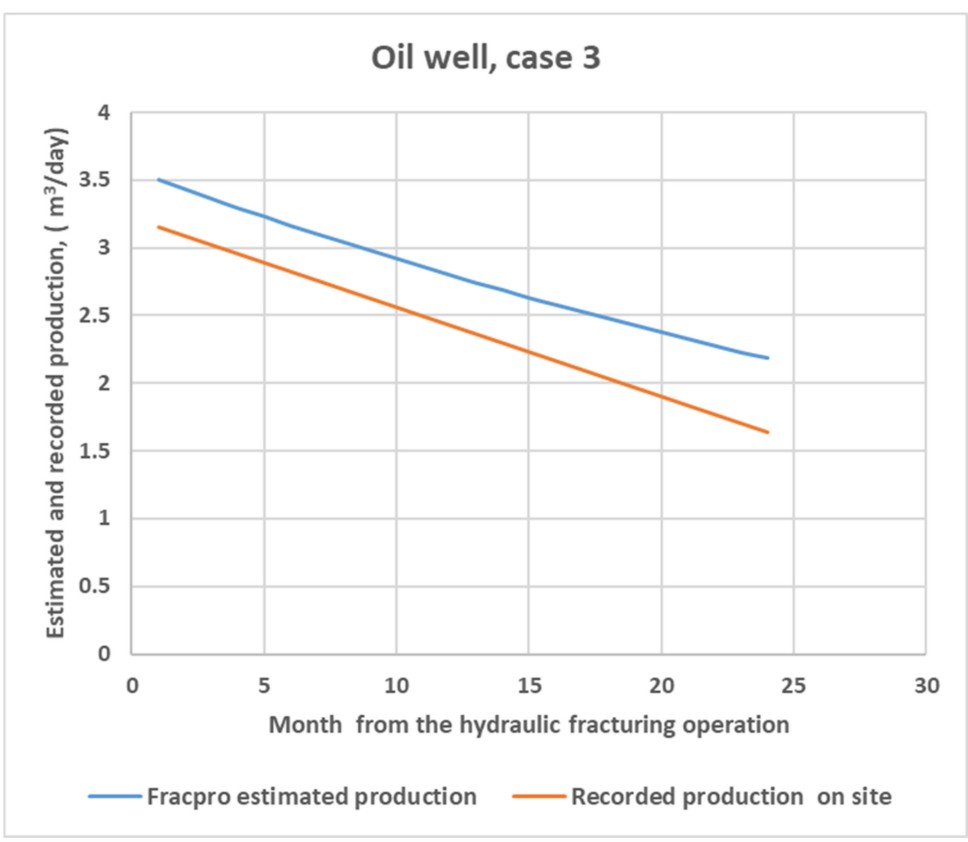

**Figure 15.** Comparison between the estimated oil production and the recorded production for 24 months. Case 3.

One aspect that deserves to be emphasized is the low costs involved in carrying out the fracturing treatment, compared to the benefits obtained from the capitalizing of the production. Recent increases in both oil and gas prices favor economic advantage. In the two examples given for a natural gas well (Case 1) and the oil well (Case 3), the cost of the fracturing operation was 0.16% and 50% of the benefits obtained in a rather short time (2 years of production). We have real production data for the two wells for which the economic analysis was carried out, as demonstrated in Figures 7 and 15. These are successful applications, but some of the wells subjected to the stimulation operation did not have very long production periods. From Figure 7, it can be noted that the natural gas well in Case 1 has been in production for 20 months, while the oil well has been in operation for the last two years in Case 3. The results in Case 3 are not as good as expected. The main reason is that we do not know all the details and we make certain approximations. This article concludes that the application of fracturing technology requires careful planning of the technology and this can, in some cases, ensure the successful stimulation of the wells. The hydraulic fracturing costs are not very high, and many applications bring important economic benefits.

**Author Contributions:** Conceptualization, I.P, I.V.G., I.G.S., F.D., G.B. and S.S.; methodology, I.P. and I.V.G.; software, I.P.; validation, I.P, I.V.G., I.G.S., F.D., G.B. and S.S.; formal analysis, I.P, I.V.G., I.G.S., F.D., G.B. and S.S.; investigation, I.P, I.V.G., I.G.S., F.D., G.B. and S.S.; resources, I.V.G., I.G.S., F.D., G.B. and S.S.; data curation, I.P, I.V.G., I.G.S., F.D., G.B. and S.S.; writing—original draft preparation, I.P, I.V.G. and I.G.S.; writing—review and editing, I.P, I.V.G., and I.G.S.; visualization, I.P, I.V.G., I.G.S., F.D., G.B. and S.S.; supervision, I.P, I.V.G., I.G.S., F.D., G.B. and S.S.; project administration, I.P, I.V.G., I.G.S., F.D., G.B. and S.S.; funding acquisition, I.P, I.V.G., I.G.S., F.D., G.B. and S.S. All authors have read and agreed to the published version of the manuscript.

**Funding:** This research received no external funding.

**Institutional Review Board Statement:** Not applicable.

**Informed Consent Statement:** Not applicable.

**Data Availability Statement:** Not applicable.

**Conflicts of Interest:** The authors declare no conflict of interest.

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
