# Peer review of "The Use of Hydraulic Fracturing in Stimulation of the Oil and Gas Wells in Romania"

_sustainability, doi:10.3390/su14095614_

Round 1

Reviewer 1 Report

The paper presents the application of the hydraulic fracturing method in Romania by three case studies. The necessity and background of hydraulic fracturing are discussed. It is suggested that the paper is published after revision.

Comments

(1) In the abstract, the background is discussed in detail, but the work done by the authors takes up only several lines, and is not presented clearly. The similar problem also appears in the conclusions. So the abstract and the conclusions needed to be rewritten.

(2) The references on fracture simulation are presented. The CZM model is frequently used for rock fracture and hydraulic fracturing, such as Acceleration of a 2D/3D finite-discrete element code for geomechanical simulations using general purpose GPU computing. Comput Geotech 100:84–96, A Novel Bilinear Constitutive Law for Cohesive Elements to Model the Fracture of Pressure‑Dependent Rocks, Rock Mechanics and Rock Engineering, 55:521–540. It is better to add the reference on CZM for fracture simulations.

(3) Some clerical error appears such as the unit m^3.

(4) In the case studies, it is better to point out the research method used, for example simulations, experiments, or the data of construction location. More details need to be given in the paper.

(5) All figures in this paper are not elegant. It is better to refine them.

Reviewer 2 Report

  • Overall, the article is too long. The 10-page introduction alone is uninteresting. I recommend shortening these passages and focusing on the case studies.
  • Many of the images in the text are hard to read and blur when enlarged on the monitor. It does not give the impression of a professional level article.
  • What exactly does figure 2 mean and where is it taken from? Or on what basis did this picture originate? What software was used?
  • I came across a strange reference to a picture in the text. In Chapter 3.1. Case 1, reference is made to Figure 9 and Figure 10 „The hydraulic fracturing program includes some stages (see Figure 9 and 10 for equipment details)“. However, these images are part of Section 3.2 Case 2 on page 20. This is confusing. The author refers to the same figure in chapter 3.2. Case 2, but with a different textual description  „The diagram of the down the hole equipment in the area of fractures is shown in Figure 9.“  Once the link is „stage-oriented“, the second time the link is „equipment-oriented“.  The title of the picture is as follows:  „Schematic of the depth assembly for the operation of frac“.
  • What does the sentence on line 624 mean? „The two 2 1/8 in production and control filters can be seen.“
  • The decimal place designation is not unified in the text. Sometimes a „comma“ is used, sometimes a „dot“.
  • The values in Table 2 are in thousands of Euros? And if not, what does the decimal point in the fourth row of the value 36,937 mean?
  • Units in graphs are usually given in brackets.

Round 2

Reviewer 1 Report

The paper has a great improvement. Still, some issues should be addressed before publication.

Comments

(1) The simulator, Fracpro, is used for the case studies in the paper. The parameters, data, etc, used in the simulations are necessary. So it is better to add the details allowed by the companies to be published in this paper.

(2) The figure number appears in a wrong order in the text, for example Figure 7 located before Figures 4, 5 and 6. Moreover, Figure 7(f) is lost.

(3) Please check whether some information is lost in Figures 4, 5, 6, 7, 8, 11, 12, 13, 15, 16, 17 and 18, and modify them. Some figures, e.g. Figure 11, can be changed to table.

Reviewer 2 Report

I can understand the authors starting from a series of studies. But sometimes fewer words capture a problem better than a lengthy description.
If you can't shorten it, I recommend to make the pictures even better and then publish.

Author Response

We thank you for your understanding and all constructive comments and suggestions.

Concerning the pictures, we redraw some of them and also we’ve switched figures 5, 6a, 11, 12a, 15, and 16 into tables 3, 4, 8, 9, 13, and 14, in order to be easier to read.